# PLAN THEN ACTION:
# HIGH-LEVEL PLANNING GUIDANCE REINFORCEMENT LEARNING FOR LLM REASONING

## ABSTRACT

Large language models (LLMs) have demonstrated remarkable reasoning abilities in complex tasks, often relying on Chain-of-Thought (CoT) reasoning. However, due to their autoregressive token-level generation, the reasoning process is largely constrained to local decision-making and lacks global planning. This limitation frequently results in redundant, incoherent, or inaccurate reasoning, which significantly degrades overall performance. Existing approaches, such as tree-based algorithms and reinforcement learning (RL), attempt to address this issue but suffer from high computational costs and often fail to produce optimal reasoning trajectories. To tackle this challenge, we propose **P**lan-**T**hen-**A**ction Enhanced Reasoning with **G**roup **R**elative **P**olicy **O**ptimization (*PTA-GRPO*), a two-stage framework designed to improve both high-level planning and fine-grained CoT reasoning. In the first stage, we leverage advanced LLMs to distill CoT into compact high-level guidance, which is then used for supervised fine-tuning (SFT). In the second stage, we introduce a guidance-aware RL method that jointly optimizes the final output and the quality of high-level guidance, thereby enhancing reasoning effectiveness. We conduct extensive experiments on multiple mathematical reasoning benchmarks, including MATH, AIME2024, AIME2025, and AMC23, across diverse base models such as Qwen2.5-7B-Instruct, Qwen3-8B, Qwen3-14B, and LLaMA3.2-3B. Experimental results demonstrate that *PTA-GRPO* consistently achieves stable and significant improvements across different models and tasks, validating its effectiveness and generalization.

## 1 INTRODUCTION

Large Language Models (LLMs) have recently demonstrated remarkable reasoning abilities across a wide range of complex tasks (Xu et al., 2025a; Plaat et al., 2024; Ke et al., 2025), such as mathematics (Zhang et al., 2024; Wu et al., 2024a; Liu et al., 2023) and programming (Jiang et al., 2024), by leveraging Chain-of-Thought (CoT) reasoning (Wei et al., 2022). Models with strong reasoning capabilities, including Qwen-3 (Yang et al., 2025) , DeepSeek-R1 (Wu et al., 2024b), Seed-1.5 thinking (Seed et al., 2025) and GPT-5 thinking (OpenAI, 2025), adopt CoT as a central mechanism to structure their reasoning processes. However, CoT decoding in LLMs is still a token-level Markov Decision Process (MDP) (Ouyang et al., 2022; Wan et al., 2025; Liu et al., 2025): the output of each token is determined by the context sequence generated previously. Under this setting, mainstream decoding is both autoregressive (each decision conditions only on the prefix) and locally greedy (it optimizes short-horizon token likelihood, e.g., via greedy/low-temperature choices). This combination preserves local consistency but offers little global planning, often yielding redundant or drifting chains of thought and propagating early mistakes across long horizons (Yao et al., 2023; Qu et al., 2025; Wan et al., 2025).

Prior work augments LLM reasoning with tree-style algorithms (Zhang et al., 2024; Yao et al., 2023; Wang et al., 2024a) such as Monte Carlo Tree Search (Zhang et al., 2024) or heuristic generation tree (Li et al., 2025) to widen exploration beyond single-path decoding. While effective in some cases, these approaches hinge on repeated external queries to the LLM, incurring substantial time and compute (Wang et al., 2024a). Crucially, they do not strengthen the model's internal reasoning: performance stems from outside search. When the model cannot verify intermediate steps, the search

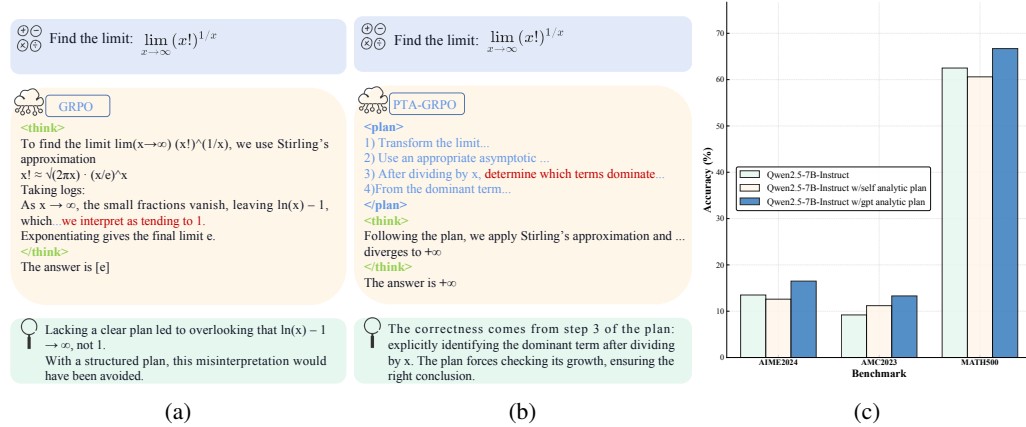

Figure 1: (a) GRPO reasoning processing. (b) *PTA-GRPO* reasoning process. (c) Impact of analytic plan. In (c), the accuracy of different reasoning modes, where Qwen2.5-7B-Instruct is considered as the base model. Green indicates the base model using CoT reasoning, yellow indicates the base model reasoning with its own self-generated analytic plan, and blue indicates the base model reasoning with an analytic plan generated by GPT-o1. More test cases of *PTA-GRPO* are shown in Appendix B.4.

simply amplifies bad branches and collapses (Feng et al., 2023). In parallel, recent works inject reflection or backtracking behaviors via RL (Wan et al., 2025; Wang et al., 2025; Gandhi et al., 2025). Such behaviors can, in principle, re-route trajectories and escape local optima (Gandhi et al., 2025). Yet when triggered on corrupted partial solutions, the model tends to reflect on its own errors, reinforcing them and drifting farther from the correct path. This occurs largely due to the absence of a global plan to guide self-reflection, leaving the model without a reliable mechanism to recover. These limitations motivate a new paradigm that improves internal planning rather than relying on external search or post-hoc self-correction.

Motivated by the way humans tackle complex problems (Kahneman, 2011), where first sketches are made and then executed, it is natural to consider whether LLM reasoning could benefit from a similar paradigm. Specifically, an LLM may first produce a compact and general analytic plan before generating a detailed CoT. Such a plan can provide concise and general global guidance (e.g., subgoal decomposition and task scheduling), and conditioning the CoT on this plan helps mitigate local myopia and reduce redundancy. However, certain weaker LLMs (e.g., Qwen-2.5-7B-Instruct (Bai et al., 2023)) lack the ability to generate high-quality analytic plans. As shown in Fig 1c, the analytic plans generated by Qwen-2.5-7B-Instruct are of insufficient quality, which actually degrades the performance of the resulting CoT and answers, whereas plans generated by the stronger model GPT-o1 lead to significant improvements. These phenomena naturally suggest that a promising direction is to enhance the analytic planning ability of LLMs, as generating high-quality analytic plans can substantially improve their reasoning performance.

To cultivate strong analytic plans, a recent advanced strategy is to exploit the advantages of Reinforcement Learning (RL), e.g., trajectory-level, non-differentiable optimization, enhancing plan quality and alignment with downstream CoT, to achieve reliable, globally guided reasoning. However, under above reasoning paradigm for analytic plan, outcome-based RL with Verifiable Rewards (RLVR) strategies (Shao et al., 2024; Yu et al., 2025; Cui et al., 2025), such as Group Relative Policy Optimization (GRPO) (Shao et al., 2024) or Decoupled Clip and Dynamic Sampling Policy Optimization (DAPO) (Yu et al., 2025), are not entirely suitable. This is because such approaches optimize only for the correctness of the final output while overlooking the quality of the analytic planning and intermediate CoT reasoning as the upper part of Fig 2. Consequently, even poorly planned and executed CoT may receive the same reward as well-structured ones, as long as both yield the correct answer. Such limitations underscore the necessity of developing new RL frameworks that can jointly optimize both the analytic planning and the detailed CoT reasoning processes.

Based on the above analysis, we propose ***PTA-GRPO*** (***p**lan-then-**a**ction enhanced reasoning with Group Relative Policy Optimization*), a novel two-stage plan-reasoning training framework designed to promote explicit higher-order planning and reasoning abilities. In the first stage, we propose a Planning-Structured Reasoning cold-start approach and leverage an advanced LLM to distill the ground-truth CoT into concise high-level guidance. Recent empirical studies (Gandhi et al., 2025;

Yue et al., 2025b; Li et al., 2025) have shown that the reasoning capabilities of pre-trained models are largely established during the initial pre-training phase, which implies that reasoning models are inherently constrained by their base models. These base models lack explicit or autonomous high-quality global planning ability. Therefore, it is necessary to cold-start and cultivate such an initial capability. To this end, the advanced LLM summarizes the CoT by extracting core concepts and generating a refined overview of the reasoning path and conclusions. This high-level guidance thinking, together with the CoT, forms a dataset for high-level guidance-based supervised fine-tuning (SFT), thereby providing a cold-start initialization for subsequent reinforcement learning. In the second stage, we propose a plan reason-guidance aware RL method based on the GRPO algorithm, which has shown strong capabilities in LLM reasoning. Unlike traditional GRPO, which rewards the model based solely on the final response, our method incorporates a sophisticated reward mechanism that evaluates the quality of the high-level guidance thinking generated during the reasoning process. This reward system not only encourages the model to generate accurate final responses but also strengthens its ability to produce effective and precise high-level guidance, thereby enhancing the model's whole reasoning ability. Our main contributions are summarized as follows:

- **A novel two-stage plan-reasoning framework:** We propose *PTA-GRPO*, a two-stage training framework, including high-level guidance planning and guidance-aware reinforcement learning, to foster explicit higher-order planning and reasoning abilities in LLMs.

- **High-level guidance as supervision signal:** In the supervised fine-tuning stage, we leverage an advanced LLM to transform raw chain-of-thought (CoT) into concise high-level guidance, which is combined with the original CoT, providing stronger initialization for reasoning.

- **Plan guidance-aware GRPO with refined reward design:** In the reinforcement learning stage, we extend GRPO with a reward mechanism that evaluates not only the correctness of the final response but also the quality of high-level guidance, significantly enhancing overall reasoning effectiveness and robustness.

## 2 PRELIMINARIES AND RELATED WORK

### 2.1 REASONING IN LARGE LANGUAGE MODELS

The reasoning of an LLM can be formalized as a token-level Markov Decision Process (MDP) (Ouyang et al., 2022; Wan et al., 2025; Liu et al., 2025), where the state is the context sequence, the action is the next token, and the policy is the model's conditional distribution. Given a question $q$, a response $\mathfrak{o} = [\mathfrak{o}^1, \ldots, \mathfrak{o}^T]$ is sampled step by step from $\pi_\theta(\cdot \mid q, \mathfrak{o}^{<t})$. Current inference typically relies on CoT, producing a reasoning chain $c$ and final answer, but this purely autoregressive process lacks global planning, often leading to redundancy and incoherence (Wan et al., 2025).

### 2.2 GROUP RELATIVE POLICY OPTIMIZATION AND ITS EXTENSIONS

GRPO (Shao et al., 2024), proposed by DeepSeek, enhances LLM reasoning without value models by sampling multiple responses per prompt and using the group average reward as a baseline. This simple mechanism has proven effective in mathematical reasoning, code generation, and QA. Subsequent variants refine GRPO from different perspectives: SRPO (Zhang et al., 2025b) reuses samples via history resampling; DAPO (Yu et al., 2025) filters extreme cases with dynamic sampling; Dr.GRPO (Liu et al., 2025) mitigates length bias; EMPO (Zhang et al., 2025a) optimizes semantic entropy directly; and SEED-GRPO (Seed et al., 2025) integrates entropy as an uncertainty measure for more conservative updates. While these methods substantially improve mathematical reasoning, they do not explicitly target higher-order reasoning abilities.

### 2.3 MOTIVATION

To address the lack of global guidance in LLM reasoning, which often leads to redundancy or off-topic reasoning, inspired by human thinking habits for complex tasks or problems (Kahneman, 2011; Kahneman & Tversky, 2013), we introduce a concise high-level plan $t$ as an outline before generating the detailed CoT $c$ and its corresponding answer. Formally, the model's output can be represented as $\mathfrak{o} = t, c$, where $t$ provides the overall problem-solving direction without involving concrete computational steps, and $c$ is then generated conditioned on both the question $q$ and the plan $t$, i.e., $c = \pi_\theta(\cdot \mid q, t)$. The CoT $c$ and its final answer are guided by the high-level plan $t$. This *plan-then-reason* mechanism equips the reasoning process with global guidance, leading to more concise, and accurate CoT generation.

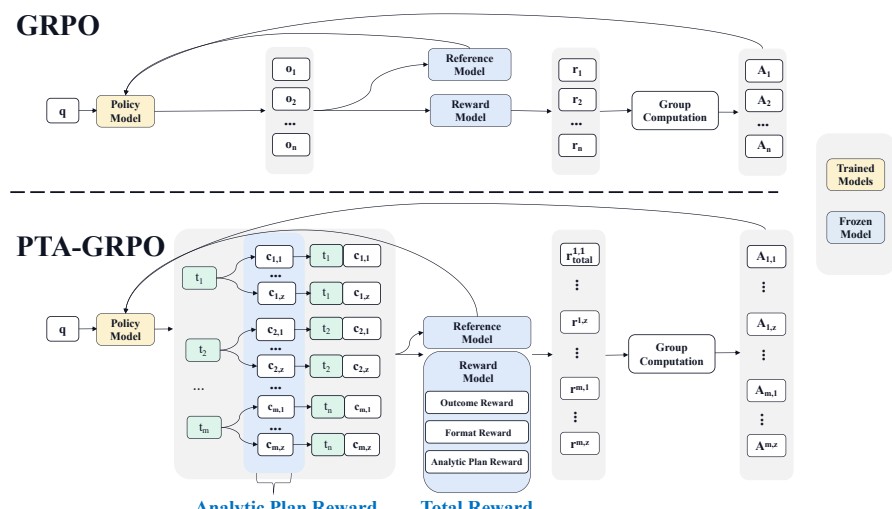

Figure 2: Comparison between GRPO and *PTA-GRPO*. It is worth noting that, to ensure a fair comparison, we keep the number of rollout responses in the RL process the same for both GRPO and PTA-GRPO.

Therefore, in GRPO optimization (the formulas are shown in Appendix B.5) in our study, the objective goes beyond simply ensuring the correctness of the answer in $\mathfrak{o}$. It also includes enhancing the quality of the high-level plan $t$, with the aim of producing $t$ more accurately and effectively. By improving $t$, the model receives structured guidance that can better direct the generation of the CoT $c$ and, consequently, the final answer. This dual focus ensures that the optimization process not only rewards correct answers but also reinforces the production of high-quality intermediate reasoning, leading to more robust and generalizable reasoning behavior.

## 3 APPROACH OF *PTA-GRPO*

In this section, we introduce the *PTA-GRPO* training framework, which consists of two key components. **(1) Plain Structured Reasoning Cold-Start (PSR-CS).** This module serves as a cold-start approach built upon supervised fine-tuning (SFT). Unlike conventional SFT datasets that contain only direct CoT and answers, we first construct a novel dataset that introduces a *general analytical plan* before detailed reasoning. This additional analytical plan provides higher-level guidance, enabling the model to abstract complex problem-solving strategies into concise forms and offering explicit guidance for answer generation. **(2) Planning Structure-Guided Reinforcement Learning (PSG-RL).** In this stage, we propose a GRPO-based Structure-Guided reinforcement learning algorithm to further enhance the structural reasoning capability of the model. The model is guided to generate general analytical content, whose quality is evaluated and converted into a reward function to determine whether it facilitates more accurate answer generation. This reward signal is then integrated into the GRPO reinforcement learning loop as an explicit optimization objective, thereby forming a closed cycle that continuously improves the effectiveness of the model's reasoning.

### 3.1 PLANNING STRUCTURED REASONING COLD-START (PSR-CS)

**Analytical-Guided SFT Dataset Construction.** For LLMs, the ability to perform reasonable planning directly affects whether they can successfully solve a problem. However, existing SFT datasets typically focus only on detailed CoT reasoning and final answers, while neglecting the importance of conducting an overall analytical plan before solving the problem. To address this gap, we propose an analytical-guided dataset, which consists of three components: the problem, a general analytical plan, and the corresponding detailed CoT reasoning with the final answer. This dataset not only injects concise and effective general analytical knowledge into LLMs to provide an overall problem-solving perspective but also trains them to transform such general plans into concrete reasoning processes, thereby enhancing their overall reasoning capabilities. Formally, we define the dataset as $D_{\text{PSR-CS}} = \{q_i, t_i, c_i\}_{i=1}^n$, which contains $n$ tuples, where each tuple comprises the problem $q_i$, the general analytical plan $t_i$, and the corresponding detailed reasoning with the final answer $c_i$. Different from directly producing a CoT (Wei et al., 2022), which may lack global guidance, SFT explicitly injects the high-level problem-solving plan $t_i$ during training, enabling the model to leverage

this global information when generating the reasoning chain. Consequently, the model effectively learns the conditional distribution $c_i = \pi_\theta(\cdot \mid q_i, t_i)$ for producing detailed reasoning and the final answer, and the SFT process further strengthens this plan-to-reasoning guidance. In our constructed dataset, the general analytical plan $t_i$ is enclosed within the <plan>...</plan> tags, which clearly distinguishes the high-level problem-solving idea. Meanwhile, the specific response $c_i$ is further structured: the chain-of-thought (CoT) is wrapped in <think>...</think>, and the final answer is wrapped in <answer>...</answer>, thereby providing a hierarchical representation of planning, reasoning, and answering. In contrast to prior approaches that require multi-turn interactions (Yao et al., 2022) to obtain and follow high-level guidance, our design integrates the plan and the subsequent reasoning–answering process into a single compact response. This unified structure enables the model to complete planning and execution in a single pass and allows RLVR to efficiently optimize the plan–action components jointly.

In practice, we sampled 10K instances from the Openthoughts (Guha et al., 2025) dataset as our base. Openthoughts is a large-scale open reasoning dataset that covers a wide range of problems along with their detailed CoT reasoning processes. We then employed the powerful open-source reasoning model Qwen3-235B (Yang et al., 2025) as the teacher model. For each instance, we input the problem $q_i$ and its detailed reasoning $c_i$ into the advanced model to generate the corresponding general analytical plan $t_i$. Through this process, we distilled general analytical knowledge from a strong LLM and injected it into our target models to enhance their overall reasoning capability.

**SFT-based Cold-Start Initialization Optimization.** At this stage, we aim to inject structured reasoning capabilities into the initial policy model $\pi_\theta$ through SFT, which serves as an effective way to expand the knowledge and abilities of LLMs (Shah et al., 2025). Specifically, we optimize the model parameters by minimizing the discrepancy between the model outputs and the reference outputs provided in the analytical-guided dataset $D_{\text{SRCS}}$, thereby enabling the model to gradually acquire structured reasoning patterns. The fine-tuning process can be formulated as:

$$\theta_{\text{SFT}} = \min_\theta \quad \mathbb{E}_{(q_i, t_i, c_i) \in \mathcal{D}_{\text{SRCS}}} \left[ \sum_{i=1}^n \log \left( \pi_\theta(t_i, c_i \mid q_i) \right) \right]. \tag{1}$$

$\theta_{\text{SFT}}$ refers to the parameter set learned through supervised fine-tuning on the analytical-guided dataset. Based on these optimized parameters, $\pi_{\theta_{\text{SFT}}}$ denotes the resulting policy model that embodies structured reasoning capabilities. By explicitly injecting high-level analytical plans before detailed CoT reasoning, the policy model is guided to generate solutions in a more systematic and interpretable manner.

## 3.2 PLAN STRUCTURE-GUIDED REINFORCEMENT LEARNING (PSG-RL)

After obtaining the policy model $\pi_{\theta_{\text{SFT}}}$ from the SFT stage, the RL phase then focuses on improving the model's planning capability and ensuring its effective execution. At this stage, we not only consider the correctness of CoT $c$ and its answer as part of the reward signal, but also evaluate the quality of the analytical plan $t$, which is incorporated as another important aspect of the reward signal.

### 3.2.1 ANALYTICAL PLAN–GUIDED REWARD AUGMENTATION IN GRPO

In *PTA-GRPO*, we design a composite reward function that integrates three aspects: the analytical planning reward ($r_{\text{analytical}}$) to encourage structured reasoning plans, the outcome accuracy reward ($r_{\text{outcome}}$) to ensure correct final results, and the structured format reward ($r_{\text{format}}$) to enforce clear and consistent output. Together, these rewards are combined into the total reward $R_{\text{total}}$, which enhances the model's planning capability, reasoning accuracy, and response reliability.

**Analytical Plan Reward.** Since directly evaluating the quality of an analytical plan $t$ is difficult in practice, we instead use computable and optimizable surrogate objectives to measure the probability that it guides a specific CoT reasoning process toward the correct answer, where a higher probability intuitively reflects a higher-quality plan. Based on this insight, we design the reward for the analytical plan $r_{\text{analytic}}$, which is defined by the probability that the analytical plan can guide a CoT reasoning process toward the correct answer. To achieve the above goal, we construct a response group $G$ through a two-step process. Given a question $q$, the policy model first samples a set of $m$ candidate analytical plans $\{t_i\}_{i=1}^m$, where $t_i \sim \pi_\theta(\cdot \mid q)$ and each analytical plans $t_i$ is a concise,

text-based outline of how to approach $q$. Then, for each analytical plan $t_j$, following (Lu et al., 2025), we resample $z$ detailed CoT $\{c_{i,k}\}_{k=1}^{z}$ under guidance of $t_i$, where each $c_{i,k}$ is drawn as $c_{i,k} \sim \pi_\theta(\cdot \mid t_i, q)$. The response group $G$ consists of $m$ analytical plans, each associated with $z$ CoT, where $G = \left\{ \{(t_i, c_{i,k})\}_{k=1}^{z} \right\}_{i=1}^{m}$. For each response from $G$ can be regarded as planning-CoT paris, and the reward $r_{\text{analytic}}$ assigned to $t_i$ is defined as the empirical accuracy of its resampled outcomes:

$$r_{\text{analytic}}(t_i) = \text{Softmax}\left( \frac{1}{z} \sum_{k=1}^{z} \mathbb{I}[\hat{y}_{i,k} = y] \right), \tag{2}$$

where $\mathbb{I}[\cdot]$ is the indicator function, $\hat{y}_{i,k}$ denotes the final expected answer extracted from $c_{i,k}$, and $y$ is the ground-truth answer of $q$. Through the policy model driven by $r_{\text{analytic}}(\cdot)$, more accurate analytic plans $t$ can be generated, thereby improving the probability of obtaining the correct prediction $\Pr(\hat{y} = y \mid t, q)$. In addition, we apply the Softmax to exponentially amplify the differences between scores, making high-scoring planning more prominent while further suppressing low-scoring ones.

In contrast to traditional RLVR (Yu et al., 2025; Feng et al., 2025), which relies solely on outcome-based supervision and cannot supervise the intermediate reasoning process, our analytic plan reward $r_{\text{analytic}}$ enables us to directly assess which intermediate reasoning trajectories are more valuable and more likely to succeed, and to assign them higher rewards accordingly. Section 3.3 shows, both theoretically and empirically, that optimizing the analytic plan reward $r_{\text{analytic}}$ increases the mutual information between $y$ and $\hat{y}$, thereby enhancing reasoning ability.

**Outcome Reward.** The outcome reward, defined as $r_{\text{outcome}}$, is a result-based terminal reward similar to GRPO, used to evaluate whether the predicted answer aligns with the ground truth. For each plan–CoT response $(t_i, c_{i,k})$, the outcome reward $r_{\text{outcome}}$ is defined as follows:

$$r_{\text{outcome}} = \begin{cases} 1, & \hat{y}_{i,k} = y, \\ 0, & \text{else}. \end{cases} \tag{3}$$

The outcome reward $r_{\text{outcome}}$ encourages the policy model to learn to follow the analytical plan $t_i$ and to develop the ability to generate answers that strive for correctness.

**Format Reward.** The format reward $r_{\text{format}}$ is designed to regulate the overall structure of the model response, ensuring both conformity to the desired format and control over the output length. It consists of two components: $r_{\text{structure}}$ and $r_{\text{length}}$. Specifically, $r_{\text{structure}}$ enforces that the policy model's response adheres to the predefined structural template, i.e., `<plan>...</plan>`, `<think>...</think>`, and `<answer>...</answer>`. Meanwhile, $r_{\text{length}}$ serves as an auxiliary reward that encourages the model to generate concise and efficient token sequences, thereby reducing redundant or uninformative content.

To provide a clearer illustration of each reward, we present its detailed formulation as follows. We begin with the format reward $r_{\text{format}}$, which is defined as:

$$r_{\text{format}} = \begin{cases} 0.2, & \text{if the response strictly follows the predefined template} \\ 0, & \text{otherwise}. \end{cases} \tag{4}$$

This function enforces a binary constraint on the output structure: a full reward is granted only when the response strictly adheres to the predefined template, thereby ensuring the consistency and parsability of the generated results.

For response length, the optimal number of tokens varies across different questions, making it difficult to predefine a fixed target length. Therefore, for all responses generated for a given question, we select the shortest correct response length as the reference length $T$, defined as:

$$T = \min\{ |\{t_i, c_{i,k}\}| \mid \hat{y}_{i,k} = y \}, \tag{5}$$

where $|\{t_i, c_{i,k}\}|$ denotes the token length of response $\{t_i, c_{i,k}\}$. Here, $T$ represents the shortest executable token length required to obtain the correct answer to a given question. It can be regarded as the optimal reference length under current knowledge, toward which other correct responses should converge in order to minimize redundancy while preserving correctness. For each response $\{t_i, c_{i,k}\} \in G$, the length reward $r_{\text{length}}$ can be expressed as:

$$r_{\text{length}}(\{t_i, c_{i,k}\}) = \alpha \cdot \exp(-\frac{|\,|\{t_i, c_{i,k}\}| - T|}{T_{\max} - T}), \tag{6}$$

where $\alpha$ is a hyperparameter, and $T_{\max}$ does not denote the maximum output length set for the policy model. The reward becomes larger as the response length approaches the reference length $T$, encouraging the model to generate concise yet correct responses.

The format reward $r_{\text{format}}$, defined as $r_{\text{format}} = r_{\text{structure}} + r_{\text{length}}$, ensures that the output not only adheres to the required format, but also guarantees the conciseness of the output response.

**Total Reward.** The above three rewards together constitute the total reward $R_{\text{total}}$ for each response as:

$$R_{\text{total}} = R_{\text{analytic}} + \beta \cdot R_{\text{outcome}} + R_{\text{format}}, \tag{7}$$

where $\beta$ represents the hyperparameter. We first obtain a total reward set $\{\{r_{total}^{i,k}\}_{i=1}^{m}\}_{k=1}^{z}$, where $r_{total}^{i,k}$ denotes the total reward of the $k$-th CoT generated under the guidance of the $i$-th analytic. Based on this reward, we compute the corresponding advantage function $A_{i,k}$ using Eq. 10, and subsequently incorporate it into the update rule in Eq. 9 to optimize the model.

Table 1: Performance comparison of different post-training methods using various base models. **Bold** is best per block.

| Method | MATH500 | AIME24 | AIME25 | AMC23 | Average |
|---|---|---|---|---|---|
| **Qwen2.5-7B-Instruct** | 62.40 | 12.24 | 3.52 | 52.75 | 32.73 |
| GRPO | 82.74 | 27.52 | 22.33 | 63.59 | 49.04 |
| DAPO | 83.92 | 28.90 | 21.25 | 67.75 | 50.45 |
| CPL (Wang et al., 2024b) | 80.27 | 24.90 | 23.27 | 66.23 | 48.64 |
| Full-Step-DPO (Xu et al., 2025b) | 81.17 | 26.49 | 20.25 | 62.53 | 47.59 |
| ORZ (Hu et al., 2025) | 83.51 | 27.44 | 22.35 | 67.59 | 50.22 |
| *PTA-GRPO* | **85.57** | **30.26** | **25.97** | **70.24** | **53.01** |
| **LLaMA3.2-3B** | 34.27 | 3.33 | 2.74 | 18.75 | 14.77 |
| GRPO | 55.19 | 16.27 | 14.22 | 38.25 | 30.98 |
| DAPO | 54.27 | 18.35 | **16.53** | 38.25 | 31.85 |
| *PTA-GRPO* | **60.25** | **20.50** | 14.27 | **40.37** | **33.85** |
| **Qwen3-8B** | 90.27 | 66.67 | 51.53 | 90.05 | 74.63 |
| GRPO | 92.93 | 68.27 | 54.23 | 91.97 | 76.85 |
| DAPO | 91.27 | 66.39 | 50.08 | 91.33 | 74.77 |
| CPL (Wang et al., 2024b) | 90.75 | 67.77 | 51.44 | 90.77 | 75.18 |
| Full-Step-DPO (Xu et al., 2025b) | 91.95 | 67.29 | 52.39 | 91.15 | 75.70 |
| ORZ (Hu et al., 2025) | 92.09 | 65.67 | 53.55 | 90.98 | 75.57 |
| *PTA-GRPO* | **93.31** | **68.88** | **54.29** | **92.29** | **77.19** |
| **Qwen3-14B** | 91.27 | 72.65 | 70.03 | 94.33 | 82.07 |
| GRPO | 90.28 | 71.29 | 71.29 | 94.92 | 81.95 |
| DAPO | 91.07 | 72.33 | 70.92 | **95.20** | 82.38 |
| *PTA-GRPO* | **91.93** | **73.90** | **71.55** | 94.97 | **83.09** |

**Advantages Compared with Conventional GRPO.** Compared with standard GRPO, which primarily relies on sparse task-level accuracy supervision, our guidance-aware *PTA-GRPO* framework introduces several critical improvements. **First**, powered by the analytic-plan reward $r_{\text{analytic}}$, the model gains the ability to *evaluate* its intermediate reasoning process, which RLVR cannot achieve with purely outcome-based signals. This mechanism drives the model to construct higher-level analytic plans and use them to guide more reliable CoT reasoning. **Second**, the outcome reward $r_{\text{outcome}}$ encourages the policy model to follow the analytic plan and enhance its reasoning capability under such structured guidance. **Third**, format reward $r_{\text{format}}$ encourages stable, standardized reasoning, pushing outputs to be both concise and correct. Together, these enhancements enable *PTA-GRPO* to achieve stronger high-level analytic planning and improved reasoning performance in complex tasks compared to standard GRPO. Besides, following (Gandhi et al., 2025), PTA-GRPO enhances LLM self-reflection in reinforcement learning by adjusting the prompt in Appendix B.4, allowing the model to correct later steps even when the initial plan is flawed; detailed examples are provided there.

## 3.3 THEORETICAL PERFORMANCE ANALYSIS

In this section, we theoretically analyze the impact of optimizing $r_{\text{analytic}}$ on the performance of the policy model on the probability of errors. Our theoretical findings are as follows.

**Theorem 3.1.** *Let $q$ denote the input question, $t$ the analytic plan, $\hat{y}$ the answer predicted by the policy model, and $y$ the ground-truth answer. With error probability $p_{error}$, it holds that:*

$$p_{error} \leq \tfrac{1}{2}\big[H(y) - I(\hat{y}, y \mid t, q)\big], \quad p_{error} = \Pr(y \neq \hat{y}),$$

*where $H(\cdot)$ denotes the entropy, and $I(\cdot)$ denotes the mutual information.*

The proof can be seen in appendix B.6. Leveraging the conclusion from (Qian et al., 2025), since $H(y)$ depends solely on the fixed distribution of the answer and is independent of the model's reasoning steps, it can therefore be regarded as a constant. In our Theorem 3.1, the upper bound of the error probability $p_{\text{error}}$ is governed by the conditional mutual information $I(\hat{y}; y \mid t, q)$, which measures the statistical dependence between the predicted output $\hat{y}$ and the true label $y$, given the auxiliary analytic plan $t$. In other words, the larger the shared information between $\hat{y}$ and $y$ under the guidance of the analytical plan $t$, the tighter the upper achievable limit on the probability of error.

**Proposition 3.2.** *Let $t_1$ and $t_2$ be any analytic plans, and let $r_{\text{analytic}}(t_1)$ and $r_{\text{analytic}}(t_2)$ denote their corresponding analytic rewards. Let $\hat{y}_1$ and $\hat{y}_2$ be the answers induced by executing $t_1$ and $t_2$, respectively. If $r_{\text{analytic}}(t_1) \geq r_{\text{analytic}}(t_2)$, it holds that*

$$H(y|\hat{y}_1, t_1, q) \leq H(y|\hat{y}_2, t_2, q). \tag{8}$$

*Remark* 3.3. By the definition of mutual information, $I(\hat{y}; y \mid q, t) = H(y \mid q, t) - H(y \mid \hat{y}, q, t)$. Note that $H(y \mid q, t)$ is solely determined by the underlying data distribution of $(q, t, y)$ and is independent of the model's prediction $\hat{y}$. Hence, $H(y \mid q, t)$ can be regarded as a constant with respect to the learning or inference process. As shown in Proposition 3.2, as the analytic plan reward $r_{\text{analytic}}$ increases, the conditional entropy $H(y \mid \hat{y}, q, t)$ decreases, which in turn implies a larger mutual information $I(y; \hat{y} \mid q, t)$ between the prediction $\hat{y}$ and the ground-truth $y$. In particular, we have $\max_t r_{\text{analytic}}(t) \iff \max_t I(y; \hat{y} \mid q, t)$. Therefore, optimizing the analytic plan reward term $r_{\text{analytic}}$ can effectively enhance the model's reasoning ability.

**Empirical analysis.** To assess the reliability of the above theory, we further examine the relationship between mutual information and the plan reward. Following (Qian et al., 2025), we use the Hilbert–Schmidt Independence Criterion (HSIC) to estimate the mutual information between the predicted answer $\hat{y}$ and the ground-truth answer $y$. As shown in Fig. 3, the plan reward and the mutual information exhibit similar increasing trends, which is consistent with our theoretical claim $\max_t r_{\text{analytic}}(t) \iff \max_t I(y; \hat{y} \mid q, t)$ and thus supports its validity.

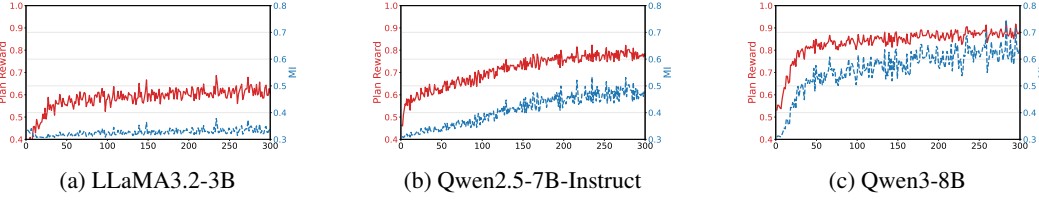

| (a) LLaMA3.2-3B | (b) Qwen2.5-7B-Instruct | (c) Qwen3-8B |

Figure 3: Trend plots of mutual information and plan reward across different models.

## 4 EXPERIMENT

**Based Models.** To evaluate *PTA-GRPO*, we adopt four base models of varying scales and series: LLaMA3.2-3B (Dubey et al., 2024), Qwen2.5-7B-Instruct (Bai et al., 2025), Qwen3-8B, and Qwen3-14B (Yang et al., 2025), enabling a comprehensive assessment of its robustness across architectures. Training details are in Section B.8. For our vision-language model (VLM), we use Qwen2.5-7B-VL (Bai et al., 2025) as the base model to further extend our experiments.

**Training Datasets and Benchmarks.** For SFT, we use 10K samples from Openthoughts (Guha et al., 2025) with injected planning knowledge (Section 3.1). For RL, we sample 14K problems from DeeMath (He et al., 2025), which offers graded difficulty and is rigorously decontaminated to avoid benchmark leakage. We evaluate our method on AIME24, AIME25, MATH500, and AMC23, and report the average accuracy over 16 independent runs. In addition, we assess it on the general-purpose multimodal datasets MMMU-Pro (Yue et al., 2025a), MMMU (Yue et al., 2024), and EMMA (Standley et al., 2023), as well as the scientific benchmark dataset MMK-12 (Meng et al., 2025).

**Baseline.** We compare *PTA-GRPO* with the base model and several RLVR like GRPO (Shao et al., 2024), and DAPO (Yu et al., 2025). In addition, we compare our method with several advanced reinforcement learning algorithms, including CPL (Wang et al., 2024b), Full-Step DPO (Xu et al., 2025b), and ORZ (Hu et al., 2025). For fairness, all methods use the same SFT and RL data (differing only in the improved SFT portion). For a fair comparison, we use the same number of sampled responses as all selected RLVR methods, so their time consumption is nearly the same.

## 4.1 PERFORMANCE OF *PTA-GRPO*

Table 2: Impact of data scale of RL on *PTA-GRPO*, where Qwen2.5-7B-Instruct is considered as base model. **Bold** is best per block.

| Data scale | MATH500 | AIME24 | AIME25 | AMC23 | Average |
|---|---|---|---|---|---|
| 4k | 82.27 | 27.22 | 21.03 | 65.22 | 48.94 |
| 8k | 83.59 | 28.23 | 22.29 | 68.29 | 50.60 |
| 11k | 84.23 | 29.33 | 24.51 | 69.37 | 51.86 |
| 14k | **85.57** | **30.26** | **25.97** | **70.24** | **53.01** |

Table 1 shows that our method (*PTA-GRPO*) consistently outperforms both the base models and other RLVR approaches across different model scales and evaluation benchmarks. For relatively weaker backbones such as Qwen2.5-7B-Instruct and LLaMA3.2-3B, *PTA-GRPO* delivers the most significant improvements, raising the average scores by over 20 points compared to the raw models and further surpassing GRPO and DAPO by clear margins.

For stronger base model such as Qwen3-8B and Qwen3-14B, the headroom for improvement is smaller, yet *PTA-GRPO* still yields consistent gains on nearly all benchmarks, setting new best average scores without any degradation. This robust general-

Table 3: Ablation analysis on *PTA-GRPO*, where Qwen2.5-7B-Instruct is considered as base model. **Bold** is best per block.

| Method | MATH500 | AIME24 | AIME25 | AMC23 | Average |
|---|---|---|---|---|---|
| *PTA-GRPO* $_{\text{w/o SFT}}$ | 79.25 | 16.25 | 12.25 | 59.22 | 41.74 |
| *PTA-GRPO* $_{\text{w/o}_{\text{format}}}$ | 85.37 | **31.23** | 24.52 | 68.25 | 52.34 |
| *PTA-GRPO* $_{\text{w/o } r_{\text{analytic}}}$ | 81.03 | 28.22 | 23.85 | 66.33 | 49.86 |
| *PTA-GRPO* | **85.57** | 30.26 | **25.97** | **70.24** | **53.01** |

ization benefits both weaker and state-of-the-art models, and we further provide significance analysis in Appendix B.2.

## 4.2 IMPACT OF RL DATA SCALING

Table 2 shows how the performance of Qwen2.5-7B-Instruct on four math benchmarks changes as the RL data scale increases from 4k to 14k. Overall, all tasks steadily improve with larger data sizes, with the average score rising from 48.94 to 53.01, indicating consistent gains from more training data. Specifically, MATH500 remains the strongest across all scales (82.27→85.57), while AIME24 and AIME25, though starting lower, achieve the largest relative improvements, particularly AIME25, which increases from 21.03 to 25.97, a gain of over 23

## 4.3 ABLATION ANALYSIS

Table 3 shows that removing SFT sharply drops the average to 41.74, underscoring its necessity; removing the format reward slightly improves AIME24 but lowers the average to 52.34; and removing the analytic reward further reduces it to 49.86, confirming its importance for reasoning quality. Overall, the full *PTA-GRPO* (with SFT, format reward, and analytic reward) attains the best performance (53.01), indicating that all components are needed for maximum stability and accuracy.

## 4.4 IMPACT OF ANALYTIC PLAN ON SFT

Table 4 compares standard SFT (w/o planning) with SFT on $\mathcal{D}_{\text{SRCS}}$ augmented by analytic plans (w/ planning). Incorporating analytic plans consistently improves all tasks and models: for Qwen2.5-7B-Instruct, the average score rises from 45.03 to 47.43 (gains of 0.67–3.59), indicating a stronger dependence of smaller models on external planning signals; for Qwen3-8B, the average improves from 75.92 to 77.46 with gains of about 1–2 points. Overall, analytic plans provide structured reasoning supervision that substantially boosts smaller models while offering steady fine-grained gains for larger ones.

Table 4: The impact of datasets containing analytic planning on SFT. **Bold** is best per block.

| Base Model | Method | MATH500 | AIME24 | AIME25 | AMC23 | Average |
|---|---|---|---|---|---|---|
| Qwen2.5-7B-Instruct | SFT w/o planning | 78.28 | 21.66 | 19.66 | 60.53 | 45.03 |
| | SFT w/ planning | **80.40** | **25.25** | **20.33** | **63.75** | **47.43** |
| Qwen3-8B | SFT w/o planning | 91.02 | 70.03 | 50.25 | 92.39 | 75.92 |
| | SFT w/ planning | **92.53** | **71.97** | **51.77** | **93.55** | **77.46** |

## 4.5 ADDITIONAL EMPIRICAL EVALUATION ON GENERALIZATION

Beyond mathematics, we also evaluate on multimodal, general-domain, and scientific benchmarks, including MMMU-Pro (Yue et al., 2025a), MMMU (Yue et al., 2024), EMMA (Standley et al., 2023), and MMK-12 (Meng et al., 2025). Using MM-EKURA (Meng et al., 2025) and SRPO (Wan et al., 2025) as baselines and following SRPO's SFT/RL data

Table 5: Comparison between PTA-GRPO and other approaches on General-Benchmark and Science Benchmark, using Qwen2.5-7B-VL as the base model.

| Method | MMMU-Pro | MMMU | EMMA | Phys | Chem | Bio |
|---|---|---|---|---|---|---|
| **Base** | 36.9 | 54.3 | 21.5 | 45.4 | 56.4 | 54.0 |
| MM-Eurek (Meng et al., 2025) | 37.6 | 55.2 | 23.5 | 45.4 | 56.4 | 54.0 |
| SRPO (Wan et al., 2025) | 42.3 | 57.1 | 29.6 | 56.2 | 65.2 | 65.2 |
| PTA-GRPO | **44.7** | **59.0** | **31.9** | **58.5** | **68.7** | **66.8** |

for cold-start and training, PTA-GRPO with Qwen2.5-7B-VL consistently outperforms the Base model, MM-EKURA, and SRPO on MMMU-Pro, MMMU, EMMA, and Phys/Chem/Bio benchmarks, achieving uniformly better metrics and stronger generalization reasoning.

## 4.6 EFFECTIVENESS ON LARGE-SCALE DATASETS WITH STRONG MODELS

As shown in Table 1, using only a small amount of data brings little improvement for strong models such as Qwen3-14B. To investigate whether this limitation is due to data scale, we expand the training set to 60K examples from the same datasets (He et al., 2025); the corresponding results are presented in Table 6 in Appendix. PTA-GRPO achieves consistently larger performance gains across all base models and mathematical benchmarks, even on the strong LLM Qwen3-14B, demonstrating that our method benefits substantially from a larger data scale and yields better overall results.

## 4.7 RESULTS OF SCALING TEST-TIME

We next examine the effectiveness of *PTA-GRPO* under multiple sampling at test time. As shown in Fig. 4, *PTA-GRPO* consistently outperforms GRPO on the AIME2025 dataset across Pass@1, Pass@4, Pass@8, and Pass@16. This demonstrates that *PTA-GRPO* maintains high precision under low-sample conditions, while further exhibiting stronger solution coverage as the number of samples increases.

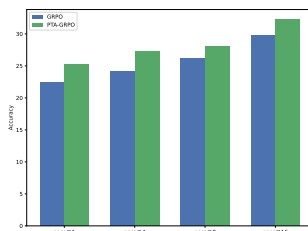

## 4.8 TRAINING DYNAMICS OF *PTA-GRPO*

Appendix B.3 Fig. 5 and Fig. 6 illustrate the training dynamics of QWEN3-8B and QWEN2.5-7B-Instruct, respectively. As shown in the figures, our method outperforms GRPO in terms of accuracy reward and response length, indicating the effectiveness of the introduced component. It is worth noting that in Fig. 5 (b), our approach achieves lower entropy compared to GRPO. This suggests that for stronger models, our method encourages the development of more

Figure 4: Effect of scaling test-time compute on AIME25 (Pass@K), with Qwen2.5-7B-Instruct as the base model.

reasonable analytic plans, enabling the model to complete a given trajectory with greater confidence and ultimately achieving higher accuracy.

## 5 CONCLUSION

We propose Plan-Guide Enhanced Reasoning with Group Relative Policy Optimization (*PTA-GRPO*), which integrates high-level planning with fine-grained reasoning to alleviate the lack of global planning in traditional CoT reasoning. Experimental results show that *PTA-GRPO* achieves stable and significant improvements across multiple mathematical reasoning benchmarks and model scales, validating its effectiveness and generalizability.

## 6 ETHICS STATEMENT

This research has been conducted in alignment with the ICLR Code of Ethics. We are committed to responsible stewardship of machine learning research, ensuring that our work advances knowledge while considering its potential societal impacts. In particular, we uphold high standards of scientific rigor, transparency, and reproducibility, and we affirm that no data has been falsified, fabricated, or misrepresented. Our study avoids harm by carefully considering possible negative consequences and by respecting privacy, fairness, and inclusiveness in the use of data and methods. All data used complies with relevant ethical approvals and license requirements, and precautions have been taken to prevent re-identification or misuse. We respect the intellectual contributions of others and provide appropriate credit where due. We believe this work contributes positively to human well-being by addressing problems of scientific and social relevance in ways that are transparent, responsible, and consistent with the principles of the ICLR Code of Ethics.

## 7 REPRODUCIBILITY STATEMENT

We have taken several steps to ensure the reproducibility of our work. The main experimental setup, including model architectures, training procedures, and evaluation metrics, is described in detail in the main paper and appendix. To facilitate reproducibility, we will release the majority of the code with an anonymous code link (shown in the Appendix) during the review process. If the paper is accepted, we commit to releasing the complete code base for all major experiments, along with detailed documentation and instructions for reproducing the reported results.

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

# A THE USE OF LARGE LANGUAGE MODELS

In preparing this manuscript, we used a Large Language Model (LLM) solely to assist with minor language polishing and improvements in readability. The LLM did not contribute to research ideation, analysis, or substantive writing. All scientific content and conclusions are entirely the responsibility of the authors.

# B APPENDIX

## B.1 EXPERIMENT RESULTS ON LARGER-SCALE DATA

Table 6: Performance comparison of RLVR methods using various base models with 60K training samples. **Bold** indicates best per block.

| Method | MATH500 | AIME24 | AIME25 | AMC23 | Average |
|---|---|---|---|---|---|
| **Qwen2.5-7B-Instruct** | 65.10 | 13.43 | 3.56 | 54.79 | 34.22 |
| GRPO | 90.65 | 29.98 | 25.00 | 70.21 | 53.96 |
| DAPO | 92.21 | 31.64 | 22.80 | 74.12 | 55.19 |
| PTA-GRPO | **94.27** | **34.38** | **28.47** | **77.78** | 58.72 |
| **Qwen3-8B-Instruct** | 91.46 | 68.31 | 52.34 | 91.26 | 75.84 |
| GRPO | **94.04** | 69.68 | 55.42 | 93.51 | 78.16 |
| DAPO | 93.12 | 68.36 | 51.61 | 92.87 | 76.49 |
| PTA-GRPO | 94.04 | **70.51** | **56.10** | **94.14** | 78.70 |
| **Qwen3-14B-Instruct** | 91.34 | 72.22 | 72.36 | 95.70 | 82.91 |
| GRPO | 92.53 | 73.68 | 72.27 | 96.14 | 83.66 |
| DAPO | 93.03 | 73.54 | 72.56 | 96.58 | 83.93 |
| PTA-GRPO | **94.38** | **75.20** | **73.49** | **97.56** | 85.15 |

## B.2 ANALYSIS OF STATISTICAL SIGNIFICANCE

Table 7: Performance comparison of RLVR methods using various base models. Results reported as mean±std over 32 seeds. **Bold** indicates best per block.

| Method | MATH500 | AIME24 | AIME25 | AMC23 | Average |
|---|---|---|---|---|---|
| **Qwen2.5-7B-Instruct** | $61.94_{\pm3.35}$ | $12.45_{\pm4.07}$ | $3.32_{\pm4.34}$ | $52.25_{\pm3.88}$ | 32.49 |
| GRPO | $82.47_{\pm2.26}$ | $27.00_{\pm2.91}$ | $22.31_{\pm3.47}$ | $64.26_{\pm3.89}$ | 49.01 |
| DAPO | $83.70_{\pm2.31}$ | $29.59_{\pm3.43}$ | $20.12_{\pm3.29}$ | $67.43_{\pm3.33}$ | 50.21 |
| PTA-GRPO (Ours) | $\mathbf{85.29_{\pm1.55}}$ | $\mathbf{30.27_{\pm2.19}}$ | $\mathbf{25.20_{\pm2.24}}$ | $\mathbf{70.46_{\pm2.83}}$ | 52.81 |
| **LLaMA3.2-3B-Instruct** | $34.24_{\pm4.35}$ | $3.37_{\pm5.91}$ | $2.10_{\pm4.74}$ | $19.24_{\pm4.53}$ | 14.74 |
| GRPO | $55.16_{\pm2.78}$ | $16.70_{\pm3.01}$ | $13.77_{\pm3.93}$ | $38.18_{\pm4.45}$ | 30.95 |
| DAPO | $54.48_{\pm3.91}$ | $18.90_{\pm3.07}$ | $\mathbf{16.46_{\pm4.01}}$ | $38.72_{\pm3.63}$ | 32.14 |
| PTA-GRPO (Ours) | $\mathbf{60.60_{\pm1.43}}$ | $\mathbf{20.75_{\pm2.86}}$ | $14.16_{\pm2.10}$ | $\mathbf{40.53_{\pm2.45}}$ | 34.01 |
| **Qwen3-8B-Instruct** | $90.09_{\pm2.09}$ | $66.89_{\pm3.19}$ | $51.22_{\pm2.80}$ | $90.38_{\pm2.83}$ | 74.65 |
| GRPO | $92.86_{\pm1.66}$ | $68.02_{\pm2.28}$ | $\mathbf{54.83_{\pm2.47}}$ | $92.33_{\pm2.59}$ | 77.01 |
| DAPO | $91.49_{\pm1.92}$ | $66.99_{\pm2.60}$ | $49.85_{\pm2.42}$ | $90.92_{\pm2.67}$ | 74.81 |
| PTA-GRPO (Ours) | $\mathbf{93.28_{\pm1.46}}$ | $\mathbf{69.92_{\pm1.77}}$ | $54.74_{\pm1.85}$ | $\mathbf{92.38_{\pm1.43}}$ | 77.58 |
| **Qwen3-14B-Instruct** | $90.53_{\pm2.14}$ | $70.61_{\pm2.92}$ | $68.55_{\pm2.42}$ | $93.65_{\pm2.82}$ | 80.84 |
| GRPO | $90.71_{\pm1.04}$ | $71.44_{\pm1.95}$ | $70.56_{\pm1.94}$ | $94.87_{\pm1.84}$ | 81.89 |
| DAPO | $90.89_{\pm1.36}$ | $72.22_{\pm1.88}$ | $71.04_{\pm2.03}$ | $95.26_{\pm1.67}$ | 82.35 |
| PTA-GRPO (Ours) | $\mathbf{92.11_{\pm1.39}}$ | $\mathbf{73.34_{\pm1.63}}$ | $\mathbf{71.63_{\pm1.80}}$ | $\mathbf{95.56_{\pm1.65}}$ | 83.16 |

To more accurately quantify the reliability of our results, we increased the number of independent runs for the remaining experiments from 16 to 32 (Table 7) and conducted t-tests for significance analysis (Table 8). As shown in Table 8, compared with the existing DAPO and GRPO methods, our approach achieves a significant improvement in performance.

## B.3 TRAINING DYNAMICS

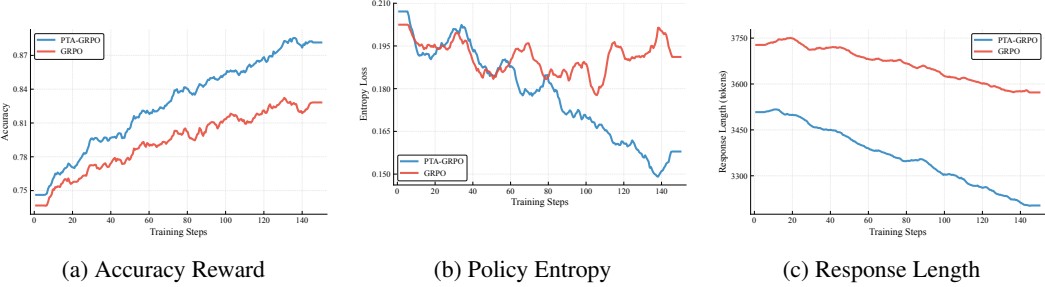

(a) Accuracy Reward          (b) Policy Entropy          (c) Response Length

Figure 5: Training Dynamics of *PTA-GRPO* with Qwen3-8B.

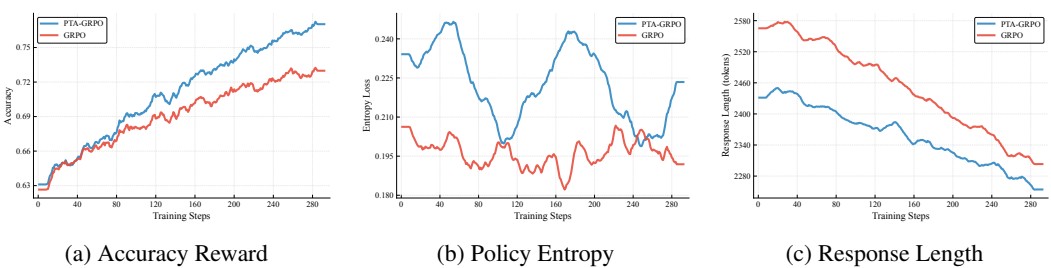

(a) Accuracy Reward          (b) Policy Entropy          (c) Response Length

Figure 6: Training Dynamics of *PTA-GRPO* with Qwen2.5-7B-Instruct.

## B.4 TEST CASE OF *PTA-GRPO*

### Generated Samples in Real Test cases of *PTA-GRPO* (Sample 1)

**Question:** Determine the interval of convergence for the series

$$1 + \frac{x}{2} + \frac{2!x^2}{3^2} + \frac{3!x^3}{4^3} + \dots$$

**\<plan\>**

1. Identify the general term of the series. 2. Apply the Ratio Test to find the radius of convergence. 3. Check the endpoints of the interval separately to determine convergence or divergence there. 4. State the final interval of convergence.

<think>

We start by writing down the general term of the series. The pattern is clear: the $n$-th term looks like

$$\frac{n!}{(n+1)^n}x^n.$$

To test for convergence, the Ratio Test is natural. Taking the ratio $a_{n+1}/a_n$, most factors cancel, and it reduces to

$$|x|\left(\frac{n+1}{n+2}\right)^{n+1}.$$

As $n \to \infty$, the power term tends to $e^{-1}$. So the limit is $|x|/e$. By the Ratio Test, convergence requires $|x|/e < 1$, which means $|x| < e$. That gives us the radius of convergence $R = e$.

Now check the endpoints. At $x = e$, the terms are $\frac{n!}{(n+1)^n}e^n$. Stirling's formula shows that $n! \sim (n/e)^n\sqrt{2\pi n}$, so the terms behave like $\sqrt{n}$ times a constant. They don't go to zero, so the series diverges. At $x = -e$, the alternating sign doesn't help because the size of the terms again grows like $\sqrt{n}$. They still don't go to zero, so the series diverges there as well. Therefore, the series converges only strictly inside the interval, and the interval of convergence is

$$(-e, e).$$

**Answer:**

$$\boxed{(-e,\ e)}$$

## Generated Samples in Real Test cases of *PTA-GRPO* (Sample 2)

**Question:** What is the least possible value of the sum $|x - 1| + |x - 1.5| + |x - 2|$?

<plan>

1. Recognize that this is a minimization problem involving sums of absolute values. 2. Recall that for a sum of absolute values of linear functions, the minimum occurs at a median of the points. 3. Identify the points: 1, 1.5, and 2. 4. Determine the median of these three points. 5. Calculate the sum at the median point to find the minimum value.

<think>

We want to minimize

$$f(x) = |x - 1| + |x - 1.5| + |x - 2|.$$

General fact: For $|x - a_1| + \cdots + |x - a_n|$, the minimum occurs when $x$ is the median of the $a_i$.

Here, the three points are $1, 1.5, 2$. Ordered: $1 < 1.5 < 2$. So the median is 1.5.

Evaluate at $x = 1.5$:

$$|1.5 - 1| + |1.5 - 1.5| + |1.5 - 2| = 0.5 + 0 + 0.5 = 1.$$

Check at neighbors:

* At $x = 1$: $0 + 0.5 + 1 = 1.5$. * At $x = 2$: $1 + 0.5 + 0 = 1.5$. * At $x = 1.25$: $0.25 + 0.25 + 0.75 = 1.25$. * At $x = 1.75$: $0.75 + 0.25 + 0.25 = 1.25$.

All larger than 1. Thus, the minimum occurs uniquely at $x = 1.5$, with value 1.

**Answer:**

$$\boxed{1}$$

## Generated Samples in Real Test cases of *PTA-GRPO* (Sample 3)

**Question:** Find the range of $p$ for which the stability condition $|\xi| \leq 1$ holds, given that

$$|\xi|^2 = 1 - 4p^2(1 - p^2)\,s^4, \quad s = \sin\left(\tfrac{\omega}{2}\right) \in [0, 1].$$

**<plan>**

1. Understand the stability condition: $|\xi| \leq 1$. 2. Substitute the given expression for $|\xi|^2$. 3. The stability condition $|\xi| \leq 1$ is equivalent to $|\xi|^2 \leq 1$. 4. Since $|\xi|^2$ is expressed as $1 - 4p^2(1 - p^2)s^4$, ensure it is non-negative ($\geq 0$) to avoid complex values. 5. Analyze the sign of $p^2(1 - p^2)$. 6. Determine the condition on $p$ such that $|\xi|^2 \in [0, 1]$ for all $s \in [0, 1]$. 7. Conclude the range of $p$.

**<think>**

We have

$$|\xi|^2 = 1 - 4p^2(1 - p^2)s^4, \quad s \in [0, 1].$$

For stability, we need $|\xi| \leq 1$, i.e., $|\xi|^2 \leq 1$. That means

$$1 - 4p^2(1 - p^2)s^4 \leq 1.$$

This simplifies to

$$-4p^2(1 - p^2)s^4 \leq 0 \quad \Rightarrow \quad p^2(1 - p^2) \geq 0.$$

Since $p^2 \geq 0$, this requires $1 - p^2 \geq 0 \implies p^2 \leq 1$. Thus $|p| \leq 1$.
Additionally, we must ensure $|\xi|^2 \geq 0$. For $|p| \leq 1$, the term $4p^2(1 - p^2)s^4 \geq 0$, so $|\xi|^2 = 1 - (\text{nonnegative}) \in [0, 1]$. This guarantees both realness and the upper bound condition.
If $|p| > 1$, then $1 - p^2 < 0$, so $p^2(1 - p^2) < 0$. In this case the subtracted term becomes negative, effectively making $|\xi|^2 > 1$ for some $s$, which breaks stability.
Hence the stability condition holds exactly when

$$p \in [-1, 1].$$

**Answer:**

$$\boxed{[-1, 1]}$$

## Example of an LLM correcting an incorrect plan through self-reflection.

**Question:** Six points $A, B, C, D, E,$ and $F$ lie in a straight line in that order. Suppose that $G$ is a point not on the line and that $AC = 26$, $BD = 22$, $CE = 31$, $DF = 33$, $AF = 73$, $CG = 40$, and $DG = 30$. Find the area of $\triangle BGE$.

**<plan>**

1. Assign coordinates to points A through F on the x-axis with A at 0. 2. Use the given segment lengths (AC, BD, CE, DF, AF) to set up equations and solve for all individual segment lengths AB, BC, CD, DE, EF. 3. Determine the coordinates of B, C, D, E, F. 4. Use distances CG and DG to locate point G by solving a system of equations from the distance formula. 5. Calculate area of triangle BGE using the shoelace formula with coordinates B, G, E. \\Wrong plan

**<think>**

Wait, I notice the plan is incorrect in its approach to finding point G. The plan says to use distances CG and DG to locate G, but this will only give us the x-coordinate of G, not both coordinates. Let me revise this. \\Revise it by LLM self-reflection

Looking more carefully at the problem, I need to find the area of triangle BGE. The points B, G, and E form a triangle where: - B and E are on the x-axis - G is somewhere in the plane The area of triangle BGE can be calculated as ½ × |BE| × height from G to line BE. Since BE is horizontal (both B and E on x-axis), the height is simply |y-coordinate of G|.

So the revised approach should be: 1. Find coordinates of B and E 2. Find the y-coordinate of G 3. Area = $\frac{1}{2} \times |BE||y_G|$.

Let me implement this revised plan.

First, let's find all segment lengths as in the original plan: Let AB = a, BC = b, CD = c, DE = d, EF = e

From AC = 26: a + b = 26 From BD = 22: b + c = 22 From CE = 31: c + d = 31 From DF = 33: d + e = 33 From AF = 73: a + b + c + d + e = 73

From a + b = 26 and a + b + c + d + e = 73, we get c + d + e = 47 From c + d = 31 and c + d + e = 47, we get e = 16 From d + e = 33 and e = 16, we get d = 17 From c + d = 31 and d = 17, we get c = 14 From b + c = 22 and c = 14, we get b = 8 From a + b = 26 and b = 8, we get a = 18

So: AB = 18, BC = 8, CD = 14, DE = 17, EF = 16

Place A at (0,0), then: B = (18, 0) C = (26, 0) D = (40, 0) E = (57, 0) F = (73, 0)

Now, to find the y-coordinate of G: We know CG = 40 and DG = 30 Let G = (x,y)

CG² = (x-26)² + y² = 1600 DG² = (x-40)² + y² = 900

Subtract: (x-26)² - (x-40)² = 700 (x² - 52x + 676) - (x² - 80x + 1600) = 700 28x - 924 = 700 28x = 1624 x = 58

Now use (x-26)² + y² = 1600: (58-26)² + y² = 1600 32² + y² = 1600 1024 + y² = 1600 y² = 576 y = 24 (taking positive)

So G = (58, 24)

Now triangle BGE: B = (18, 0), E = (57, 0), G = (58, 24) BE = 57 - 18 = 39 Height = 24 (y-coordinate of G)

Area = ½ × 39 × 24 = 468

**Answer:**

$$\boxed{468}$$

---

**PTA-GRPO prompt**

```
{{ content | trim }}
Begin by outlining a high-level plan wrapped in <plan></plan>
tags.
- This plan should capture only the major phases, strategic
choices, and conditional branches.
- Avoid low-level steps, calculations, or detailed reasoning
here.  Next, reason step by step within <think></think>.
- During reasoning, critically evaluate the initial plan.  If
you find any errors, inconsistencies, or improvements needed,
```

```
   revise your plan mentally and continue reasoning based on the
   revised plan.
   - Explicitly state if you are revising the plan and describe
   the changes.
   - This is your detailed chain-of-thought:  work through
   assumptions, intermediate steps, and logical derivations until
   the solution is reached.
   Finally, provide the final answer enclosed within

   \boxed{}
```

## B.5 GROUP RELATIVE POLICY OPTIMIZATION (GRPO)

Group Relative Policy Optimization (GRPO) is a state-of-the-art Reinforcement Learning with Verifiable Rewards (RLVR) algorithm that simplifies Proximal Policy Optimization (PPO) (Schulman et al., 2017) by removing the need for a value model to estimate the baseline advantage, and has demonstrated remarkable success in enhancing the reasoning abilities of LLM. Formally, let $Q$ denote the set of questions, $\pi_{\theta_{old}}$ be the current policy model, and $\{\mathfrak{o}_i\}_{i=1}^N$ represent a collection of $N$ candidate responses sampled for a question $q \in Q$. We also define $\pi_{\theta_{ref}}$ as a fixed reference model. The training objective of GRPO is expressed as:

$$
J_{\text{GRPO}}(\theta) = \mathbb{E}_{q \sim Q, \{\mathfrak{o}_i\}_{i=1}^N \sim \pi_{\theta_{old}}}
$$

$$
\left[ \frac{1}{N} \sum_{i=1}^N \sum_{t=1}^{|\mathfrak{o}_i|} \min \left( \frac{\pi_\theta(\mathfrak{o}_i^t|q)}{\pi_{\theta_{old}}(\mathfrak{o}_i^t|q)} A_i, \text{clip} \left( \frac{\pi_\theta(\mathfrak{o}_i^t|q)}{\pi_{\theta_{old}}(\mathfrak{o}_i^t|q)}, 1-\epsilon, 1+\epsilon \right) A_i \right) - \beta D_{\text{KL}}(\pi_\theta \| \pi_{\text{ref}}) \right] \quad (9)
$$

where $\epsilon$ controls the clipping range and $\beta$ weights the KL regularization term. The normalized advantage $A_i$ assigned to each response $\mathfrak{o}_i$ is computed from group-based rewards:

$$
A_i = \frac{r_i - \mu}{\sigma}, \quad \text{with } \mu = \frac{1}{N} \sum_{j=1}^N r_j, \quad \sigma = \sqrt{\frac{1}{N} \sum_{j=1}^N (r_j - \mu)^2}, \quad (10)
$$

where $\{r_1, r_2, \ldots, r_N\}$ are the scalar rewards associated with the response group $\{\mathfrak{o}_i\}_{i=1}^N$.

In GRPO, each response $\mathfrak{o} \in \{\mathfrak{o}_i\}_{i=1}^N$ consists of a CoT $c$ together with its final answer. As noted in Section 2.1, token-level MDPs lack global planning and often yield redundant steps, while GRPO rewards $r$ corresponding to $\mathfrak{o}$ focus only on final answer correctness, overlooking reasoning quality and enabling reward hacking through superficial or verbose CoTs.

## B.6 THEORETICAL PROOF

*Proof.* Following the framework of (Qian et al., 2025), for a fixed $(q, t)$, the conditional error rate is

$$
p_e(q, t) = 1 - \max_{y'} \Pr(y = y' \mid q, t).
$$

For the binary distribution $(p, 1 - p)$, it is known that

$$
\min(p, 1 - p) \leq \tfrac{1}{2} H_b(p),
$$

where $H_b(p) = -p \log p - (1 - p) \log(1 - p)$ is the binary entropy. This can be generalized to the $m$-class case.

**Lemma B.1.** *Let $(p_1, \ldots, p_m)$ be a probability distribution, and let $p_{\max} = \max_i p_i$. Then*

$$
1 - p_{\max} \leq \tfrac{1}{2} H(p_1, \ldots, p_m).
$$

*Proof by induction.* **Base case** $m = 2$. This is exactly the binary inequality.

*Induction step.* Suppose the inequality holds for $(m-1)$ classes. Consider an $m$-class distribution with maximum element $p_1$, and let $s = 1 - p_1$. Merge the last two categories into one, obtaining an $(m-1)$-class distribution $\tilde{\boldsymbol{p}}$. By the grouping property of Shannon entropy,

$$H(p_1, \ldots, p_{m-2}, p_{m-1}, p_m) = H(\tilde{\boldsymbol{p}}) + (p_{m-1} + p_m)H_b\left(\frac{p_{m-1}}{p_{m-1}+p_m}\right) \geq H(\tilde{\boldsymbol{p}}).$$

By the induction hypothesis,

$$s = 1 - p_1 \leq \tfrac{1}{2}H(\tilde{\boldsymbol{p}}) \leq \tfrac{1}{2}H(p_1, \ldots, p_m).$$

Thus the lemma holds for all $m$. $\qquad\square$

For the conditional distribution $\Pr(y \mid q, t)$, the lemma implies

$$p_e(q, t) \leq \tfrac{1}{2}H(y \mid q, t).$$

Taking expectation over $(q, t)$,

$$p_{\text{error}} = \mathbb{E}_{q,t}[p_e(q, t)] \leq \tfrac{1}{2}H(y \mid q, t).$$

By the chain rule,

$$I(\hat{y}; y \mid q, t) = H(y \mid q, t) - H(y \mid \hat{y}, q, t),$$

which implies

$$H(y \mid q, t) \geq H(y \mid \hat{y}, q, t).$$

Also,

$$H(y \mid q, t) = H(y) - I(y; q, t).$$

Combining these gives

$$p_{\text{error}} \leq \tfrac{1}{2}H(y \mid q, t) \leq \tfrac{1}{2}\big[H(y) - I(y; \hat{y} \mid q, t)\big].$$

The theorem is proved. $\qquad\square$

### B.7 PROOF OF PROPOSITION 3.2

*Proof.* Let $p_i = \Pr(\hat{y}_i = y \mid t_i, q)$ denote the probability that the final answer generated under plan $t_i$ is correct. The analytic reward $r_{\text{analytic}}(t_i)$ is a monotonic function of the empirical estimate of $p_i$. Therefore, the condition $r_{\text{analytic}}(t_1) \geq r_{\text{analytic}}(t_2)$ implies:

$$p_1 \geq p_2. \tag{1}$$

The conditional entropy $H(y \mid \hat{y}_i, t_i, q)$ measures the remaining uncertainty about the true answer $y$ after observing the predicted answer $\hat{y}_i$ generated under plan $t_i$. We decompose this entropy by conditioning on whether $\hat{y}_i$ is correct:

$$H(y \mid \hat{y}_i, t_i, q) = \Pr(\hat{y}_i = y \mid t_i, q) \cdot H(y \mid \hat{y}_i = y, t_i, q)$$
$$+ \Pr(\hat{y}_i \neq y \mid t_i, q) \cdot \mathbb{E}[H(y \mid \hat{y}_i, t_i, q) \mid \hat{y}_i \neq y].$$

If the predicted answer $\hat{y}_i$ is correct (i.e., the event $\hat{y}_i = y$ occurs), then the posterior distribution $\Pr(y \mid \hat{y}_i, t_i, q)$ collapses to a point mass on the value $\hat{y}_i$, resulting in zero conditional entropy:

$$H(y \mid \hat{y}_i = y, t_i, q) = 0.$$

Let $C_i = \mathbb{E}[H(y \mid \hat{y}_i, t_i, q) \mid \hat{y}_i \neq y]$ denote the expected conditional entropy when the predicted answer is wrong. Substituting into the equation above yields:

$$H(y \mid \hat{y}_i, t_i, q) = p_i \cdot 0 + (1 - p_i) \cdot C_i = (1 - p_i)C_i. \tag{2}$$

We now compare the entropies for the two plans:

$$H(y \mid \hat{y}_1, t_1, q) = (1 - p_1)C_1, \quad H(y \mid \hat{y}_2, t_2, q) = (1 - p_2)C_2.$$

From (1), we have $1 - p_1 \leq 1 - p_2$.

We now introduce the core assumption: a plan with a higher analytic reward provides more informative guidance, leading to a posterior distribution over $y$ that is more concentrated even when the predicted answer is incorrect. Formally, this means:

$$C_1 \leq C_2. \tag{3}$$

This is reasonable because a high-quality plan constrains the reasoning path more effectively, reducing the set of plausible wrong answers and resulting in lower uncertainty upon observing an incorrect prediction.

Since $(1 - p_1) \leq (1 - p_2)$ and $C_1 \leq C_2$, and all terms are non-negative, it follows that:

$$(1 - p_1)C_1 \leq (1 - p_2)C_2.$$

Therefore, by (2):
$$H(y \mid \hat{y}_1, t_1, q) \leq H(y \mid \hat{y}_2, t_2, q),$$

which completes the proof. $\qquad\qquad\qquad\qquad\qquad\qquad\qquad\qquad\qquad\qquad\qquad\square$

## B.8 EXPERIMENTAL PARAMETER SETUP

We conducted all experiments on eight H200 GPUs. In the supervised fine-tuning (SFT) stage, we trained Qwen2.5-7B-Instruct for 3 epochs. In the reinforcement learning (RL) stage, we adopted the GRPO algorithm, with a global batch size of 128 and a micro batch size of 4 per GPU. During rollout, the model generated 12 samples per step, including 4 analytic plans, each corresponding to 3 Chain-of-Thought (CoT) reasoning trajectories. For generation, we set temperature = 1.0 and top-p = 1.0, while for validation we used temperature = 0.6, top-p = 0.95. The number of RL training steps was configured as follows: LLaMA3.2-3B and Qwen2.5-7B-Instruct were trained for 350 steps, Qwen3-8B for 150 steps, and Qwen3-14B for 50 steps, with other hyperparameters kept the same across models. In addition, the learning rate (`lr`) was set to $1.0 \times 10^{-6}$, the weight decay (`weight_decay`) was $1.0 \times 10^{-2}$, the optimizer was `adamw` (choices: `adamw` or `adamw_bf16`), the learning-rate warmup ratio (`lr_warmup_ratio`) was 0. For all Qwen3-8b, max token is 4.5k and for Qwen2.5-7B-Instrct , the max token is 3.5K.

Table 8: Pairwise significance tests between PTA-GRPO and each baseline (Instruct, GRPO, DAPO). Each row shows: improvement $\Delta$ (percentage points), $p$-value, and significance level. $*$, $**$, and $***$ denote $p < 0.05$, $p < 0.01$, and $p < 0.001$; "ns" = not significant.

| Model | Baseline | Task | $\Delta$ (pt) | $p$ / sig |
|---|---|---|---|---|
| Qwen2.5-7B | Instruct | MATH500 | +23.35 | $0.0000^{***}$ |
| | Instruct | AIME24 | +17.82 | $0.0000^{***}$ |
| | Instruct | AIME25 | +21.88 | $0.0000^{***}$ |
| | Instruct | AMC23 | +18.21 | $0.0000^{***}$ |
| | GRPO | MATH500 | +2.82 | $0.0000^{***}$ |
| | GRPO | AIME24 | +3.27 | $0.0000^{***}$ |
| | GRPO | AIME25 | +2.88 | $0.0000^{***}$ |
| | GRPO | AMC23 | +6.20 | $0.0000^{***}$ |
| | DAPO | MATH500 | +1.59 | $0.0003^{***}$ |
| | DAPO | AIME24 | +0.68 | $0.2454^{ns}$ |
| | DAPO | AIME25 | +5.08 | $0.0000^{***}$ |
| | DAPO | AMC23 | +3.03 | $0.0000^{***}$ |
| LLaMA3.2-3B | Instruct | MATH500 | +26.36 | $0.0000^{***}$ |
| | Instruct | AIME24 | +17.38 | $0.0000^{***}$ |
| | Instruct | AIME25 | +12.06 | $0.0000^{***}$ |
| | Instruct | AMC23 | +21.29 | $0.0000^{***}$ |
| | GRPO | MATH500 | +5.44 | $0.0000^{***}$ |
| | GRPO | AIME24 | +4.05 | $0.0000^{***}$ |
| | GRPO | AIME25 | +0.39 | $0.4651^{ns}$ |
| | GRPO | AMC23 | +2.34 | $0.0001^{***}$ |
| | DAPO | MATH500 | +6.12 | $0.0000^{***}$ |
| | DAPO | AIME24 | +1.86 | $0.0006^{***}$ |
| | DAPO | AIME25 | -2.29 | $0.0000^{***}$ |
| | DAPO | AMC23 | +1.81 | $0.0028^{**}$ |
| Qwen3-8B | Instruct | MATH500 | +3.19 | $0.0000^{***}$ |
| | Instruct | AIME24 | +3.03 | $0.0000^{***}$ |
| | Instruct | AIME25 | +3.52 | $0.0000^{***}$ |
| | Instruct | AMC23 | +2.00 | $0.0023^{**}$ |
| | GRPO | MATH500 | +0.42 | $0.3477^{ns}$ |
| | GRPO | AIME24 | +1.90 | $0.0024^{**}$ |
| | GRPO | AIME25 | -0.10 | $0.8679^{ns}$ |
| | GRPO | AMC23 | +0.05 | $0.9307^{ns}$ |
| | DAPO | MATH500 | +1.79 | $0.0001^{***}$ |
| | DAPO | AIME24 | +2.93 | $0.0000^{***}$ |
| | DAPO | AIME25 | +4.88 | $0.0000^{***}$ |
| | DAPO | AMC23 | +1.46 | $0.0162^{*}$ |
| Qwen3-14B | Instruct | MATH500 | +1.58 | $0.0003^{***}$ |
| | Instruct | AIME24 | +2.73 | $0.0000^{***}$ |
| | Instruct | AIME25 | +3.08 | $0.0000^{***}$ |
| | Instruct | AMC23 | +1.90 | $0.0034^{**}$ |
| | GRPO | MATH500 | +1.40 | $0.0000^{***}$ |
| | GRPO | AIME24 | +1.90 | $0.0000^{***}$ |
| | GRPO | AIME25 | +1.07 | $0.0003^{***}$ |
| | GRPO | AMC23 | +0.68 | $0.0266^{*}$ |
| | DAPO | MATH500 | +1.22 | $0.0000^{***}$ |
| | DAPO | AIME24 | +1.12 | $0.0002^{***}$ |
| | DAPO | AIME25 | +0.59 | $0.0656^{ns}$ |
| | DAPO | AMC23 | +0.29 | $0.3203^{ns}$ |

