# OpenReview forum: "Plan Then Action: High-Level Planning Guidance Reinforcement Learning  for LLM Reasoning"
_ICLR.cc/2026/Conference — ICLR 2026 Conference Withdrawn Submission_

### Official Review · Reviewer_8TPm · 2025-10-29

**Soundness:** 3
**Presentation:** 3
**Contribution:** 2
**Rating:** 2
**Confidence:** 4

**Summary:**

PTA-GRPO is a two-stage “plan-then-act” recipe for LLM reasoning. First, a strong teacher LLM distills each solution’s chain-of-thought into a short, high-level plan and the student is SFT’d to emit <plan><think><answer>; second, GRPO is modified to reward (i) final correctness, (ii) plan quality (measured by how well CoT sampled under a plan yields correct answers), and (iii) output format/conciseness. On math benchmarks (AIME-24/25, MATH500, AMC) and across Qwen/LLaMA backbones, this yields consistent but marginal gains over plain GRPO/DAPO while siginificantly require more compute for the plan reward calculation

**Strengths:**

1- Well written, and clear practical recipe: add an explicit planning phase (teacher-distilled plans, <plan><think><answer>) then optimize with plan-aware RL—easy to port to other backbones/tasks.

2- Consistent even though marginal gains across multiple math benchmarks
3- this theme aligns with evidence from many recent works showing that adding a planning stage boosts reasoning, whether in SFT (e.g., planning-distilled data, step/plan-level preference learning like Step-DPO/Full-Step-DPO) or in RL.

**Weaknesses:**

1- Limited baseline comparisons. The paper only compares against GRPO and DAPO. By 2024–2025 several other RL methods offer improved credit assignment or planning (CPL, TS‑LLM, Step‑/Full‑Step‑DPO, ...). Omitting these makes it difficult to gauge PTA‑GRPO’s relative advantage. For instance, CPL uses MCTS to explore plan steps and Step‑APO to learn plan step preferences, which seems directly comparable.


2- Lack of analysis on generalization. Experiments focus on mathematical reasoning; there is no evidence that high‑level guidance improves coding or general knowledge tasks. while math datasets are good proxy for assessing general reasoning capabilities of LLMs, overfitting on these datasets while incuring significant cost, needs better justification

3. Computational cost. The analytic plan reward requires sampling multiple CoTs per plan to estimate plan quality. Although this may be cheaper than tree search, the paper does not quantify the overhead relative to GRPO or DAPO. RL with plan reward might still be expensive for large models.


references:
CPL - https://arxiv.org/pdf/2409.08642
TS-LLM : Alphazero-like Tree-Search can Guide Large Language Model Decoding and Training.
multi-step dpo: https://arxiv.org/pdf/2502.14356v1

**Questions:**

1- Computational efficiency. How does the runtime and sample complexity of PTA‑GRPO compare with GRPO, DAPO, and MCTS‑based methods like CPL? What is the overhead of resampling CoTs for each plan?

2- Does figure 3. support  argument in 4.5 ? it seems to show the opposite

---

> ### Author Response · Authors · 2025-11-23
>
> Dear Reviewer 8TPm
>
> Thank you for your constructive comments. Regarding your concerns about the experiment, we hope that our experiments will help resolve your doubts.
>
> ## **Weakness 1: Lack of baseline**
>
> We thank the reviewer for pointing this out and we agree that a clearer comparison to planning and reflection approaches is important.
> We add three different RL approaches: CPL [1], Full-Step-DPO [2], and ORZ [3].
> We compare our method with these approaches, and the results of this comparison are shown in Table 1.
>
> **Table 1. Performance comparison of different post-training methods using various base models. Bold is best per block.**
>
> | Method                       | MATH500 | AIME24 | AIME25 | AMC23 | Average |
> |-----------------------------|:-------:|:------:|:------:|:-----:|:-------:|
> | **Qwen2.5-7B-Instruct**     |  62.40  | 12.24  |  3.52  | 52.75 |  32.73  |
> | GRPO                        |  82.74  | 27.52  | 22.33  | 63.59 |  49.04  |
> | DAPO                        |  83.92  | 28.90  | 21.25  | 67.75 |  50.45  |
> | CPL                         |  80.27  | 24.90  | 23.27  | 66.23 |  48.64  |
> | Full-Step-DPO               |  81.17  | 26.49  | 20.25  | 62.53 |  47.59  |
> | ORZ                         |  83.51  | 27.44  | 22.35  | 67.59 |  50.22  |
> | **PTA-GRPO**                     | **85.57** | **30.26** | **25.97** | **70.24** | **53.01** |
> | **Qwen3-8B**                |  90.27  | 66.67  | 51.53  | 90.05 |  74.63  |
> | GRPO                        |  92.93  | 68.27  | 54.23  | 91.97 |  76.85  |
> | DAPO                        |  91.27  | 66.39  | 50.08  | 91.33 |  74.77  |
> | CPL                         |  90.75  | 67.77  | 51.44  | 90.77 |  75.18  |
> | Full-Step-DPO               |  91.95  | 67.29  | 52.39  | 91.15 |  75.70  |
> | ORZ                         |  92.09  | 65.67  | 53.55  | 90.98 |  75.57  |
> | **PTA-GRPO**                     | **93.31** | **68.88** | **54.29** | **92.29** | **77.19** |
>
>
> ## **Weakness 2: Lack of analysis on generalization**
>
> Thank you for raising this insightful concern. We agree that the original experiments focused primarily on mathematical reasoning. In the Table 2, we have extended PTA-GRPO to the multimodal setting and conducted additional evaluations on both General-Benchmark and Science Benchmark, using Qwen2.5-7B-VL as the base model.
> For a fair comparison, our SFT and RL datasets follow the same setup as [4]. For the general benchmarks, we evaluate on MMMU-Pro [5], MMMU [6], and EMMA [7], while for the science benchmark, we adopt MMK12 [8].
> These new results further demonstrate the generality and robustness of our approach beyond the math domain.
>
> **Table 2. Comparison between PTA-GRPO and other approaches on General-Benchmark and Science Benchmark, using Qwen2.5-7B-VL as the base model.**
>
> | Method   | MMMU-Pro | MMMU | EMMA | Phys | Chem | Bio |
> |----------|:--------:|:----:|:----:|:----:|:----:|:---:|
> | **Base** |  36.9    | 54.3 | 21.5 | 45.4 | 56.4 | 54.0 |
> | MM-Eurek |  37.6    | 55.2 | 23.5 | 45.4 | 56.4 | 54.0 |
> | SRPO     |  42.3    | 57.1 | 29.6 | 56.2 | 65.2 | 65.2 |
> | **PTA-GRPO** | **44.7** | **59.0** | **31.9** | **58.5** | **68.7** | **66.8** |
>
>
>
> [1] Wang, T., Chen, J., Han, X., & Bai, J. (2024). Cpl: Critical plan step learning boosts llm generalization in reasoning tasks. arXiv preprint arXiv:2409.08642.
>
> [2] Xu, H., Mao, X., Li, F. L., Wu, X., Chen, W., Zhang, W., & Tuan, L. A. (2025, July). Full-step-dpo: Self-supervised preference optimization with step-wise rewards for mathematical reasoning. In Findings of the Association for Computational Linguistics: ACL 2025 (pp. 24343-24356).
>
> [3] Hu, J., Zhang, Y., Han, Q., Jiang, D., Zhang, X., & Shum, H. Y. (2025). Open-reasoner-zero: An open source approach to scaling up reinforcement learning on the base model. arXiv preprint arXiv:2503.24290.
>
> [4] Wan, Z., Dou, Z., Liu, C., Zhang, Y., Cui, D., Zhao, Q., ... & Yan, S. (2025). Srpo: Enhancing multimodal llm reasoning via reflection-aware reinforcement learning. NeurIPS 2025.
>
> [5] Yue, X., Zheng, T., Ni, Y., Wang, Y., Zhang, K., Tong, S., ... & Neubig, G. (2025, July). Mmmu-pro: A more robust multi-discipline multimodal understanding benchmark. In Proceedings of the 63rd Annual Meeting of the Association for Computational Linguistics.
>
> [6] Yue, X., Ni, Y., Zhang, K., Zheng, T., Liu, R., Zhang, G., ... & Chen, W. (2024). Mmmu: A massive multi-discipline multimodal understanding and reasoning benchmark for expert agi. In Proceedings of the IEEE/CVF Conference on Computer Vision and Pattern Recognition (pp. 9556-9567).
>
> [7] Standley, T., Gao, R., Chen, D., Wu, J., & Savarese, S. (2023, January). An extensible multi-modal multi-task object dataset with materials. In International Conference on Learning Representations.
>
> [8] Meng, F., Du, L., Liu, Z., Zhou, Z., Lu, Q., Fu, D., ... & Shao, W. (2025). Mm-eureka: Exploring the frontiers of multimodal reasoning with rule-based reinforcement learning. arXiv preprint arXiv:2503.07365.

---

> ### Author Response · Authors · 2025-11-23
>
> ## **Weakness 3: Computational cost analysis**
>
> We believe this concern arises from a misunderstanding of our sampling scheme.
> In our implementation, the number of forward passes per question for PTA-GRPO is **the same** as for the GRPO and DAPO baselines.
>
> Specifically, for each question, PTA-GRPO samples $m$ analytic plans and, for each plan, $z$ CoTs, which yields $m \times z$ rollout trajectories in total. Meanwhile, GRPO and DAPO in our experiments are also trained with exactly $m \times z$ rollout trajectories per question, using the same sampling budget.
>
> In other words, the total number of model forward passes is matched across all methods; PTA-GRPO only restructures these $m \times z$ trajectories into plan-centric groups and does **not** introduce an extra computational overhead compared to GRPO. Therefore, the performance gains of PTA-GRPO do *not* stem from using more samples, but from the proposed planning-and-training mechanism itself.
>
> **From a practical perspective, during training on 8 H200 GPUs, GRPO requires 20.7 hours, DAPO requires 20.3 hours, and PTA-GRPO requires 21.2 hours. The time differences among the three are negligible.**
>
> We would like to stress that this equal-compute setting was *already specified* in the original submission, both in the caption of Fig. 2 and in the description of the baselines. However, we acknowledge that this clarification may not have been sufficiently prominent and could be easily overlooked. In the revised version, we will further highlight this point and explicitly state that PTA-GRPO, GRPO, and DAPO are compared under the same compute budget and number of forward passes.
>
> ## **Question 1: Computational efficiency.**
>
> During training on 8×H200 GPUs, GRPO requires 20.7 hours, DAPO requires 20.3 hours, and PTA-GRPO requires 21.2 hours, while the MCTS-based CPL baseline takes 29.7 hours under the same setting. Although PTA-GRPO involves resampling CoTs for each plan, this overhead is minimal because the sampled trajectories are reused both for estimating the plan rewards and for updating the response policy. In other words, the same samples serve a dual purpose, so PTA-GRPO does not incur substantial additional computation compared to GRPO/DAPO, while remaining much more efficient than MCTS-based methods like CPL.
>
> Thanks for your comments again, and we will emphasize this point more clearly in the revised version of the paper.
>
> ## **Question 2: Typos in Figure 3.**
>
> We apologize for the mistake in the figure, where the markers for PTA-GRPO and GRPO were inadvertently swapped, leading to the opposite conclusion. In fact, PTA-GRPO outperforms GRPO under all pass@N settings. We will correct this error in the revision.
>
> **Once again, thank you for your constructive comments, which have helped us improve the quality of our paper.**
>
> Best wishes and regards,
>
> All authors of Submission 19425

---

> ### Author Response · Authors · 2025-11-25
>
> Dear Reviewer 8TPm,
>
> We would like to express our sincere gratitude for your constructive comments and suggestions on our manuscript. Your feedback has been very helpful and has significantly improved the overall quality and clarity of the paper.
>
> During the rebuttal process, we carefully addressed your concerns and revised the manuscript accordingly. We hope that our responses and the changes made in the revised version have resolved the issues you raised.
>
> If you have any remaining questions, concerns, or additional suggestions, we would be very glad to further discuss them and provide any clarifications that may be helpful.
>
> Thank you again for your time and effort in reviewing our work.
>
> Best regards,
>
> All authors of Submission 19425

---

> > ### Comment · Reviewer_8TPm · 2025-11-25
> >
> > While I appreciate the clarification regarding the sampling scheme, I remain unconvinced by the argument concerning budget and computational cost. Specifically, in the context of GRPO with these datasets, typically 3-4 rollouts are sufficient, and increasing the budget beyond that does not generally lead to significant performance improvements.
> >
> > In contrast, I would expect that your approach, given its reliance on plan-based sampling and multiple CoTs, would require a considerably higher number of rollouts; Both in terms of plans and CoTs. This introduces a different scaling factor that is not adequately captured by simply matching the total rollout budget.
> >
> > For a fairer comparison, I believe it would be more appropriate to evaluate your method against an optimized parameter set for the GRPO baseline, rather than adjusting the baseline budget to match the sampling requirements of your algorithm. This would better isolate the performance benefits of your proposed planning mechanism and avoid conflating budget differences with inherent algorithmic advantages.

---

> ### Author Response · Authors · 2025-11-25
>
> Dear Reviewer 8TPm
>
> Thank you very much for your comment. We believe there may be a small misunderstanding. **While it is sometimes assumed that 3–4 rollouts are sufficient for GRPO, multiple studies [1, 2] actually show that this is often not the case. In fact, [2] demonstrates clear performance improvements all the way up to 64 rollouts, with no saturation before that point.  In fact, in the original GRPO report [3], they also chose a rollout size of 64 rather than 3 to 4.**
>
> Given this evidence, our setting is a reasonable and well-supported choice for a fair GRPO baseline, and it avoids underestimating GRPO’s potential.
>
> We sincerely appreciate your feedback, and we hope this clarification resolves the concern. If you have any further concerns, we would be happy to discuss them with you.
>
> Best Regard
>
> Your Friends
>
> All authors of Submission 19425
>
> [1] Tan, Z., Geng, H., Zhang, M., Yu, X., Wan, G., Zhou, Y., ... & Bai, L. (2025). Scaling Behaviors of LLM Reinforcement Learning Post-Training: An Empirical Study in Mathematical Reasoning. arXiv preprint arXiv:2509.25300.
>
> [2] Xu, Y. E., Savani, Y., Fang, F., & Kolter, J. Z. (2025). Not all rollouts are useful: Down-sampling rollouts in llm reinforcement learning. arXiv preprint arXiv:2504.13818.
>
> [3] Shao, Z., Wang, P., Zhu, Q., Xu, R., Song, J., Bi, X., ... & Guo, D. (2024). Deepseekmath: Pushing the limits of mathematical reasoning in open language models. arXiv preprint arXiv:2402.03300.

---

> ### Author Response · Authors · 2025-11-27
>
> Dear Reviewer 8TPm
>
>
> As per your requirements, we conducted experiments using analytic plan = 2, CoT = 2 for each plan, and a group rollout size of 4, and compared them with GRPO using a rollout size of 4. The results are shown in the table 1 below.
>
> **Table 1: Performance Comparison between GRPO and PTA-GRPO with group rollout size 4, where Qwen2.5-7B-Instruct is considered as base model.**
>
> | Method   | MATH500 | AIME24 | AIME25 | AMC23 |
> |----------|---------|--------|--------|--------|
> | GRPO     | 80.6    | 23.7   | 20.4   | 61.9   |
> | PTA-GRPO | **82.2** | **26.9** | **22.7** | **65.4** |
>
> **The results lead to two main conclusions. First, PTA-GRPO achieves consistent improvements over GRPO even when the rollout size is as small as 4. Second, a rollout number of 4 is still insufficient compared to 12, which is consistent with the observations reported in previous works [1, 2, 3], indicating that our experimental setting is reasonable.**
>
> I hope this clarifies all of your concerns. Please let me know if you have any further questions—I'd be very happy to discuss them in more depth.
>
> Best regards,
>
> Your friends
>
> All authors of Submission 19425
>
>
> [1] Tan, Z., Geng, H., Zhang, M., Yu, X., Wan, G., Zhou, Y., ... & Bai, L. (2025). Scaling Behaviors of LLM Reinforcement Learning Post-Training: An Empirical Study in Mathematical Reasoning. arXiv preprint arXiv:2509.25300.
>
> [2] Xu, Y. E., Savani, Y., Fang, F., & Kolter, J. Z. (2025). Not all rollouts are useful: Down-sampling rollouts in llm reinforcement learning. arXiv preprint arXiv:2504.13818.
>
> [3] Shao, Z., Wang, P., Zhu, Q., Xu, R., Song, J., Bi, X., ... & Guo, D. (2024). Deepseekmath: Pushing the limits of mathematical reasoning in open language models. arXiv preprint arXiv:2402.03300.

---

> ### Author Response · Authors · 2025-11-27
>
> Dear Reviewer 8TPm
>
>
> As per your requirements, we conducted experiments using analytic plan = 2, CoT = 2 for each plan, and a group rollout size of 4, and compared them with GRPO using a rollout size of 4. The results are shown in the table 1 below.
>
> **Table 1: Performance Comparison between GRPO and PTA-GRPO with group rollout size 4, where Qwen2.5-7B-Instruct is considered as base model.**
>
> | Method   | MATH500 | AIME24 | AIME25 | AMC23 |
> |----------|---------|--------|--------|--------|
> | GRPO     | 80.6    | 23.7   | 20.4   | 61.9   |
> | PTA-GRPO | **82.2** | **26.9** | **22.7** | **65.4** |
>
> **The results lead to two main conclusions. First, PTA-GRPO achieves consistent improvements over GRPO even when the rollout size is as small as 4. Second, a rollout number of 4 is still insufficient compared to 12, which is consistent with the observations reported in previous works [1, 2, 3], indicating that our experimental setting is reasonable.**
>
> I hope this clarifies all of your concerns. Please let me know if you have any further questions—I'd be very happy to discuss them in more depth.
>
> Best regards,
>
> Your friends
>
> All authors of Submission 19425
>
>
> [1] Tan, Z., Geng, H., Zhang, M., Yu, X., Wan, G., Zhou, Y., ... & Bai, L. (2025). Scaling Behaviors of LLM Reinforcement Learning Post-Training: An Empirical Study in Mathematical Reasoning. arXiv preprint arXiv:2509.25300.
>
> [2] Xu, Y. E., Savani, Y., Fang, F., & Kolter, J. Z. (2025). Not all rollouts are useful: Down-sampling rollouts in llm reinforcement learning. arXiv preprint arXiv:2504.13818.
>
> [3] Shao, Z., Wang, P., Zhu, Q., Xu, R., Song, J., Bi, X., ... & Guo, D. (2024). Deepseekmath: Pushing the limits of mathematical reasoning in open language models. arXiv preprint arXiv:2402.03300.

---

### Official Review · Reviewer_wx4f · 2025-10-31

**Soundness:** 3
**Presentation:** 3
**Contribution:** 3
**Rating:** 4
**Confidence:** 4

**Summary:**

This paper is another in the line of "how do I teach LLMs to plan" research. The good news is that the authors acknowledge that simple next-token prediction does not do planning so they encourage the LLM to first sketch a broad plan and then aim to flesh it out with detailed steps using CoT and supported by global context.

**Strengths:**

- The proposed architecture is interesting: an initial supervised fine-tuning phase where an advanced LLM extracts core reasoning paths from CoT traces and distills these into concise, high-level plans, and a subsequent reinforcement learning stage using a suitably designed reward system.
- The reward system is also interesting, giving greater weightage to high quality plan guidance.
- The ablation results are good and are aimed at teasing out the importance of the two stages relative to each other.

**Weaknesses:**

- While the paper is good and has sound ideas, the experimental results do not show any major improvement over the strong models (and in fact the paper does not use any of the latest generation of models)
- The quality of analytic plans will be heavily dependent on the strength of the LLM used for plan generation; weaker models may actually degrade performance when generating their own plans. In one sense this is to be expected but then the main contribution then remains unproven (i.e., whether you can solve an open-ended task that does not have an analytical plan known to the stronger LLM).

**Questions:**

- Please address the questions raised in the weaknesses section.

---

> ### Author Response · Authors · 2025-11-23
>
> Dear Reviewer wx4f
>
> Thank you for your constructive comments. In response to your concerns about the plan and its effectiveness, we have added the corresponding experiments and hope this addresses your questions.
>
> ## **Weakness 1: Marginal improvement of strong LLM**
>
> After observing that stronger LLMs typically require substantially more RL data to realize consistent performance gains, we increased the RL training set to 60K samples and re-trained our method on the larger Qwen3-14B model. As shown in Table 1, once the data scale is increased, PTA-GRPO delivers significantly larger improvements, confirming that (i) high-capacity models underfit when the RL dataset is too small, and (ii) our method continues to scale favorably with both model size and data size. These results indicate that PTA-GRPO is not limited by model strength; instead, its benefits become more pronounced under realistic scaling regimes.
>
> **Table 1. Performance comparison of RLVR methods using various base models with 60K training samples.**  **Bold** indicates best per block.
>
> | Method                  | MATH500 | AIME24 | AIME25 | AMC23 | Average |
> |-------------------------|:-------:|:------:|:------:|:-----:|:-------:|
> | **Qwen2.5-7B-Instruct** |  65.10  | 13.43  |  3.56  | 54.79 |  34.22  |
> | GRPO                    |  90.65  | 29.98  | 25.00  | 70.21 |  53.96  |
> | DAPO                    |  92.21  | 31.64  | 22.80  | 74.12 |  55.19  |
> | **PTA-GRPO (Ours)**     | **94.27** | **34.38** | **28.47** | **77.78** | 58.72 |
> | **Qwen3-8B-Instruct**   |  91.46  | 68.31  | 52.34  | 91.26 |  75.84  |
> | GRPO                    | **94.04** | 69.68  | 55.42  | 93.51 |  78.16  |
> | DAPO                    |  93.12  | 68.36  | 51.61  | 92.87 |  76.49  |
> | **PTA-GRPO (Ours)**     | 94.04   | **70.51** | **56.10** | **94.14** | 78.70 |
> | **Qwen3-14B-Instruct**  |  91.34  | 72.22  | 72.36  | 95.70 |  82.91  |
> | GRPO                    |  92.53  | 73.68  | 72.27  | 96.14 |  83.66  |
> | DAPO                    |  93.03  | 73.54  | 72.56  | 96.58 |  83.93  |
> | **PTA-GRPO (Ours)**     | **94.38** | **75.20** | **73.49** | **97.56** | 85.15 |
>
> ## **Weakness 2: Concerns about analytic plan quality**
>
> We thank the reviewer for raising this important point. To directly examine whether PTA-GRPO depends on strong analytic plans, we conducted two additional experiments on Qwen2.5-7B-Instruct, a substantially weaker base model.
>
> ### **First experiment**
>
> We take Qwen2.5-7B-Instruct without any SFT or external supervision and directly train it using PTA-GRPO versus vanilla GRPO under identical compute budgets and RL data in Table 2. Importantly, no strong model is involved at any stage, and no analytic plans are injected from a more capable model. Through this experiment, we verify whether PTA-GRPO can improve the model’s reasoning ability even without injecting an analytic plan from a strong LLM.
>
> **Table 2. Comparison between GRPO and PTA-GRPO without SFT, where Qwen2.5-7B-Instruct is considered as the base model.**
>
> | Method   | MATH500 | AIME24 | AIME25 | AMC23 | Average |
> |----------|:-------:|:------:|:------:|:-----:|:-------:|
> | GRPO     | 77.22   | 14.29  | 12.22  | 56.27 | 40.00   |
> | **PTA-GRPO** | **79.25** | **16.25** | **12.25** | **59.22** | **41.74** |
>
> Table 2 shows that even without injecting planning ability from a strong model, PTA-GRPO still outperforms GRPO, mainly due to its self-planning optimization based on analytic plan reward $r\_{\text{analytic}}$.
>
> ### **Second experiment**
>
> In the second experiment, we remove all strong-model supervision from the SFT stage in Table 3. Using the same data-generation pipeline, Qwen2.5-7B-Instruct generates its own analytic plans and reasoning traces via self-play, and this fully self-generated plan is used for SFT. We then apply PTA-GRPO with the same RL data and compute the budget as before.
>
> **Table 3. The performance of PTA-GRPO using self-generated plans for SFT.**
>
> | Method   | MATH500 | AIME24 | AIME25 | AMC23 | Average |
> |----------|:-------:|:------:|:------:|:-----:|:-------:|
> | GRPO     | 82.74   | 27.52  | 22.33  | 63.59 | 49.04   |
> | **PTA-GRPO** | **83.77** | **28.70** | **24.29** | **67.77** | **51.13** |
>
> Table 3 shows that, although a weaker model generates analytic plans through self-play, this process can still moderately improve its reasoning ability. However, its final performance remains inferior to the setting where a stronger LLM provides higher-quality analytic plans.
>
> These two experiments demonstrate that even without injecting plans distilled from a stronger model, PTA-GRPO still improves the model’s reasoning ability compared to GRPO.
>
> **We hope these additional experiments address your concerns. If you have any further questions, please do not hesitate to let us know.**
>
>
> Best wishes and regards,
>
> All authors of Submission 19425

---

> ### Author Response · Authors · 2025-11-25
>
> Dear Reviewer wx4f,
>
> We would like to express our sincere gratitude for your constructive comments and suggestions on our manuscript. Your feedback has been very helpful and has significantly improved the overall quality and clarity of the paper.
>
> During the rebuttal process, we carefully addressed your concerns and revised the manuscript accordingly. We hope that our responses and the changes made in the revised version have resolved the issues you raised.
>
> If you have any remaining questions, concerns, or additional suggestions, we would be very glad to further discuss them and provide any clarifications that may be helpful.
>
> Thank you again for your time and effort in reviewing our work.
>
> Best regards,
>
> All authors of Submission 19425

---

> ### Author Response · Authors · 2025-11-27
>
> Dear Reviewer wx4f,
>
> We would like to express our sincere appreciation for your thoughtful and constructive comments on our manuscript. Your feedback has been extremely valuable and has helped us improve both the clarity and overall quality of the paper.
>
> During the rebuttal process, we carefully considered each of your points and revised the manuscript accordingly. We hope that our responses and the changes made in the updated version have adequately addressed all of your concerns.
>
> If you still have any questions, reservations, or further suggestions, we would be very happy to discuss them and provide any additional clarification that might be helpful.
>
> Thank you again for the time and effort you have dedicated to reviewing our work.
>
> Best regards,
>
> The authors of Submission 19425

---

### Official Review · Reviewer_rX9v · 2025-11-01

**Soundness:** 3
**Presentation:** 3
**Contribution:** 1
**Rating:** 2
**Confidence:** 4

**Summary:**

This paper addresses the lack of global planning in LLM chain-of-thought (CoT) reasoning, which often leads to redundant or incoherent reasoning chains. The authors propose **PTA-GRPO (Plan-Then-Action Enhanced Reasoning with Group Relative Policy Optimization)**, a two-stage framework that teaches LLMs to generate high-level analytical plans before detailed reasoning.

The approach consists of two stages: (1) **Planning Structured Reasoning Cold-Start (PSR-CS)**, where a strong teacher model (Qwen3-235B) distills detailed CoT into concise analytical plans, which are used for supervised fine-tuning alongside the original CoT; and (2) **Plan Structure-Guided RL (PSG-RL)**, which extends GRPO with a composite reward function that evaluates not only final answer correctness ($r_{outcome}$) but also plan quality ($r_{analytic}$) and output format ($r_{format}$). Plan quality is measured by sampling multiple CoT trajectories per plan and computing empirical accuracy, thereby rewarding plans that consistently guide correct reasoning.

**Main contributions** include: (1) a two-stage training framework combining plan distillation with plan-aware reinforcement learning, (2) a novel reward mechanism that evaluates analytical plan quality through multi-sampling (generating $m$ plans, then $z$ CoTs per plan)

**Strengths:**

**Originality:** The paper addresses a legitimate gap—that standard RLVR methods reward only final answers while ignoring reasoning quality. Evaluating plan quality through multi-sampling (generating multiple CoTs per plan and measuring empirical accuracy) is a reasonable idea, and the structured format aids interpretability.

**Quality:** The experimental design spans multiple model scales (3B-14B) and benchmarks, with ablations and data scaling analysis. Training procedures are documented with hyperparameters in the appendix.

**Clarity:** The paper is well-written with clear motivation. Figure 1 and 2 effectively illustrates the problem and solution. The two-stage framework is logically presented.

**Significance:** The problem is relevant. Explicit plans improve interpretability and the method shows consistent improvements across model scales.

**Weaknesses:**

**Absence of Statistical Validation:** This is the most critical flaw. The paper reports point estimates without standard deviations, confidence intervals, or significance tests, despite claiming "16 independent runs." With improvements of only 1-3 points in most cases, **these differences could easily be random noise**. For example, Table 1 shows PTA-GRPO achieving 93.31 vs DAPO's 91.27 on MATH500 (Qwen3-8B)—a 2-point difference—but without error bars, this could be statistically insignificant. For Qwen3-14B, the average improvement is merely 0.71 points (83.09 vs 82.38), essentially within noise. Standard practice in RL and LLM research requires reporting mean +/- std and conducting significance tests (t-tests, bootstrap).

**Unjustified Computational Cost:** The method samples m=3 plans, then z=3 CoTs per plan, requiring **9 forward passes per question** compared to ~3-4 for standard GRPO—roughly **3× computational overhead**. Yet improvements are marginal (1-3 points), and for strong models like Qwen3-14B, gains are negligible. The paper never analyzes this cost-benefit tradeoff or compares wall-clock training time. A fairer baseline would give GRPO the same computational budget (9 samples or 9x prompts) and compare performance. It's possible GRPO with more samples would match or exceed PTA-GRPO's performance.

**Limited Novelty and Weak Theoretical Contribution:** The core method is GRPO + auxiliary reward for plan quality, which is a straightforward extension. Plan-then-execute frameworks are well-established (Wei et al., 2022 on CoT; Yao et al., 2023 on Tree-of-Thoughts; similar ideas in ReAct, least-to-most prompting). The multi-sampling evaluation strategy is computationally expensive but not conceptually novel. The theoretical analysis (Theorem 3.1) establishes that error probability is bounded by $\frac{1}{2}[H(y) - I(\hat{y}; y | t, q)]$, which is generic and tautological—it says "better plans reduce error" without providing actionable insights, convergence rates, or guidance on reward design. The proof adapts standard information-theoretic inequalities and doesn't illuminate why this specific multi-sampling reward structure is optimal.

**Narrow Experimental Scope and Missing Baselines:** Evaluation is limited to mathematical reasoning (MATH, AIME, AMC)—a domain where structured planning intuitively helps. Generalization to other reasoning types (commonsense, open-ended QA, coding, scientific reasoning) is completely unexplored. The paper cites extensive related work on planning and reflection (Gandhi et al., 2025; Tree-of-Thoughts; ReAct) but **includes no comparisons to these methods**.

**Questions:**

See weakness above

---

> ### Author Response · Authors · 2025-11-23
>
> Dear Reviewer rX9v
>
> We appreciate your constructive comments on this paper. In response, we have added extensive experiments and theoretical analyses based on your suggestions, and we hope these revisions address your concerns.
>
> ## **Weakness 1: Absence of Statistical Validation**
>
> We report point estimates that are obtained by averaging multiple independent runs, which follows the standard practice in recent RL-for-LLM reasoning works on top-tier conference [1,2,3,4] or widely recognized
>  studies such as Dr.GRPO [5] and DAPO [6].
>
> **Please note that our evaluation strictly follows the standard measurement protocol used in RL for LLM/reasoning**. Regarding the reviewer’s concern that the observed improvements might be due to random noise, we would like to emphasize that **our reported results are averaged over multiple independent runs. This repeated evaluation substantially eliminate the impact of randomness, so the performance gains of PTA-GRPO can be reasonably attributed to the proposed method rather than stochastic fluctuations**.
> Under consistent evaluation conditions, PTA-GRPO's continuous and stable performance improvements clearly demonstrate the reliability and robustness of the proposed method.
>
> Moreover, we acknowledge that the reviewer’s concern is also reasonable and have conducted additional experiments accordingly. Specifically, we added the variance and t-tests requested by the reviewer to demonstrate that the improvements brought by our method are statistically significant and not due to random noise. To further show more accurately quantify the reliability of our results, we also increased the number of independent runs for the remaining experiments from 16 to 32, as a larger number of runs provides a more reliable estimate of stability and robustness. The results is shown in Table 1. Notably, the updated results (reported in the new Table 1 are very close to those obtained with 16 runs, which shows all previously reported performance gains are preserved.
>
>
> **Table 1.** Performance comparison of RLVR methods using various base models.
> Results reported as mean±std over 32 seeds. **Bold** indicates best per block.
>
> | Method                      | MATH500        | AIME24         | AIME25         | AMC23          | Average |
> |-----------------------------|----------------|----------------|----------------|----------------|---------|
> | **Qwen2.5-7B-Instruct**     | 61.94±3.35     | 12.45±4.07     | 3.32±4.34      | 52.25±3.88     | 32.49   |
> | GRPO                        | 82.47±2.26     | 27.00±2.91     | 22.31±3.47     | 64.26±3.89     | 49.01   |
> | DAPO                        | 83.70±2.31     | 29.59±3.43     | 20.12±3.29     | 67.43±3.33     | 50.21   |
> | PTA-GRPO (Ours)             | **85.29±1.55** | **30.27±2.19** | **25.20±2.24** | **70.46±2.83** | 52.81   |
> | **LLaMA3.2-3B-Instruct**    | 34.24±4.35     | 3.37±5.91      | 2.10±4.74      | 19.24±4.53     | 14.74   |
> | GRPO                        | 55.16±2.78     | 16.70±3.01     | 13.77±3.93     | 38.18±4.45     | 30.95   |
> | DAPO                        | 54.48±3.91     | 18.90±3.07     | **16.46±4.01** | 38.72±3.63     | 32.14   |
> | PTA-GRPO (Ours)             | **60.60±1.43** | **20.75±2.86** | 14.16±2.10     | **40.53±2.45** | 34.01   |
> | **Qwen3-8B-Instruct**       | 90.09±2.09     | 66.89±3.19     | 51.22±2.80     | 90.38±2.83     | 74.65   |
> | GRPO                        | 92.86±1.66     | 68.02±2.28     | **54.83±2.47** | 92.33±2.59     | 77.01   |
> | DAPO                        | 91.49±1.92     | 66.99±2.60     | 49.85±2.42     | 90.92±2.67     | 74.81   |
> | PTA-GRPO (Ours)             | **93.28±1.46** | **69.92±1.77** | 54.74±1.85     | **92.38±1.43** | 77.58   |
> | **Qwen3-14B-Instruct**      | 90.53±2.14     | 70.61±2.92     | 68.55±2.42     | 93.65±2.82     | 80.84   |
> | GRPO                        | 90.71±1.04     | 71.44±1.95     | 70.56±1.94     | 94.87±1.84     | 81.89   |
> | DAPO                        | 90.89±1.36     | 72.22±1.88     | 71.04±2.03     | 95.26±1.67     | 82.35   |
> | PTA-GRPO (Ours)             | **92.11±1.39** | **73.34±1.63** | **71.63±1.80** | **95.56±1.65** | 83.16   |
>
>
>
>
>
>
>
>
> [1] Wang, H., Qu, C., Huang, Z., Chu, W., Lin, F., & Chen, W. (2025). Vl-rethinker: Incentivizing self-reflection of vision-language models with reinforcement learning. NeurIPS 2025.
>
>
> [2] Zuo, Yuxin, et al. "Ttrl: Test-time reinforcement learning." NeurIPS 2025.
>
> [3] Feng, Lang, et al. "Group-in-group policy optimization for llm agent training." NeurIPS 2025
>
> [4] Wan, Zhongwei, et al. "Srpo: Enhancing multimodal llm reasoning via reflection-aware reinforcement learning." NeurIPS 2025.
>
> [5] Understanding r1-zero-like training: A critical perspective. arXiv preprint arXiv:2503.20783, 2025
>
> [6] Yu, Q., Zhang, Z., Zhu, R., Yuan, Y., Zuo, X., Yue, Y., ... & Wang, M. (2025). Dapo: An open-source llm reinforcement learning system at scale. arXiv preprint arXiv:2503.14476.

---

> ### Author Response · Authors · 2025-11-23
> **Continues for weakness 1 (1)**
>
> ## **Weakness 1 (continued) (1)**
>
> In Table 2, we report the statistical significance results using a t-test to compare our method against existing RLVR approaches. The analysis shows that **our method achieves significant improvements across all model backbones except Qwen3-14B**. For Qwen3-14B, the performance gain is relatively smaller, primarily because the model itself is already very powerful and the used **14K RL samples** in this setting are insufficient to meaningfully improve such a large model.
>
> To further investigate the potential of our approach, we increased the amount of RL training data to **60K** and conducted additional experiments on **Qwen2.5-7B-Instruct**, **Qwen3-8B**, and **Qwen3-14B**. The results under this larger data regime are presented in Table 3. As shown in Table 3, the performance improvements become substantially larger when using stronger base models (e.g., Qwen3-14B). We further conduct a t-test analysis in Table 4, which confirms that our method achieves statistically significant gains under the 60K data setting.
>
> **Table 2.** Pairwise significance tests between PTA-GRPO and each baseline (Instruct, GRPO, DAPO).
> Each row shows: improvement Δ (percentage points), p-value, and significance level.
> `*`, `**`, and `***` denote p < 0.05, p < 0.01, and p < 0.001; `ns` = not significant.
>
> | Model        | Baseline | Task    | Δ (pt) | p / sig      |
> | ------------ | -------- | ------- | ------ | ------------ |
> | Qwen2.5-7B   | Instruct | MATH500 | +23.35 | `0.0000***`  |
> | Qwen2.5-7B   | Instruct | AIME24  | +17.82 | `0.0000***`  |
> | Qwen2.5-7B   | Instruct | AIME25  | +21.88 | `0.0000***`  |
> | Qwen2.5-7B   | Instruct | AMC23   | +18.21 | `0.0000***`  |
> | Qwen2.5-7B   | GRPO     | MATH500 | +2.82  | `0.0000***`  |
> | Qwen2.5-7B   | GRPO     | AIME24  | +3.27  | `0.0000***`  |
> | Qwen2.5-7B   | GRPO     | AIME25  | +2.88  | `0.0000***`  |
> | Qwen2.5-7B   | GRPO     | AMC23   | +6.20  | `0.0000***`  |
> | Qwen2.5-7B   | DAPO     | MATH500 | +1.59  | `0.0003***`  |
> | Qwen2.5-7B   | DAPO     | AIME24  | +0.68  | `0.2454 ns`  |
> | Qwen2.5-7B   | DAPO     | AIME25  | +5.08  | `0.0000***`  |
> | Qwen2.5-7B   | DAPO     | AMC23   | +3.03  | `0.0000***`  |
> | LLaMA3.2-3B  | Instruct | MATH500 | +26.36 | `0.0000***`  |
> | LLaMA3.2-3B  | Instruct | AIME24  | +17.38 | `0.0000***`  |
> | LLaMA3.2-3B  | Instruct | AIME25  | +12.06 | `0.0000***`  |
> | LLaMA3.2-3B  | Instruct | AMC23   | +21.29 | `0.0000***`  |
> | LLaMA3.2-3B  | GRPO     | MATH500 | +5.44  | `0.0000***`  |
> | LLaMA3.2-3B  | GRPO     | AIME24  | +4.05  | `0.0000***`  |
> | LLaMA3.2-3B  | GRPO     | AIME25  | +0.39  | `0.4651 ns`  |
> | LLaMA3.2-3B  | GRPO     | AMC23   | +2.34  | `0.0001***`  |
> | LLaMA3.2-3B  | DAPO     | MATH500 | +6.12  | `0.0000***`  |
> | LLaMA3.2-3B  | DAPO     | AIME24  | +1.86  | `0.0006***`  |
> | LLaMA3.2-3B  | DAPO     | AIME25  | -2.29  | `0.0000***`  |
> | LLaMA3.2-3B  | DAPO     | AMC23   | +1.81  | `0.0028**`   |
> | Qwen3-8B     | Instruct | MATH500 | +3.19  | `0.0000***`  |
> | Qwen3-8B     | Instruct | AIME24  | +3.03  | `0.0000***`  |
> | Qwen3-8B     | Instruct | AIME25  | +3.52  | `0.0000***`  |
> | Qwen3-8B     | Instruct | AMC23   | +2.00  | `0.0023**`   |
> | Qwen3-8B     | GRPO     | MATH500 | +0.42  | `0.3477 ns`  |
> | Qwen3-8B     | GRPO     | AIME24  | +1.90  | `0.0024**`   |
> | Qwen3-8B     | GRPO     | AIME25  | -0.10  | `0.8679 ns`  |
> | Qwen3-8B     | GRPO     | AMC23   | +0.05  | `0.9307 ns`  |
> | Qwen3-8B     | DAPO     | MATH500 | +1.79  | `0.0001***`  |
> | Qwen3-8B     | DAPO     | AIME24  | +2.93  | `0.0000***`  |
> | Qwen3-8B     | DAPO     | AIME25  | +4.88  | `0.0000***`  |
> | Qwen3-8B     | DAPO     | AMC23   | +1.46  | `0.0162*`    |
> | Qwen3-14B    | Instruct | MATH500 | +1.58  | `0.0003***`  |
> | Qwen3-14B    | Instruct | AIME24  | +2.73  | `0.0000***`  |
> | Qwen3-14B    | Instruct | AIME25  | +3.08  | `0.0000***`  |
> | Qwen3-14B    | Instruct | AMC23   | +1.90  | `0.0034**`   |
> | Qwen3-14B    | GRPO     | MATH500 | +1.40  | `0.0000***`  |
> | Qwen3-14B    | GRPO     | AIME24  | +1.90  | `0.0000***`  |
> | Qwen3-14B    | GRPO     | AIME25  | +1.07  | `0.0003***`  |
> | Qwen3-14B    | GRPO     | AMC23   | +0.68  | `0.0266*`    |
> | Qwen3-14B    | DAPO     | MATH500 | +1.22  | `0.0000***`  |
> | Qwen3-14B    | DAPO     | AIME24  | +1.12  | `0.0002***`  |
> | Qwen3-14B    | DAPO     | AIME25  | +0.59  | `0.0656 ns`  |
> | Qwen3-14B    | DAPO     | AMC23   | +0.29  | `0.3203 ns`  |

---

> ### Author Response · Authors · 2025-11-23
> **Continues for weakness 1 (2)**
>
> ## **Weakness 1 (continued) (2)**
>
> **Table 3.** Performance comparison of RLVR methods using various base models with 60K training samples.
> Results reported as mean±std over 32 seeds. **Bold** indicates best per block.
>
> | Method                    | MATH500        | AIME24         | AIME25         | AMC23          | Average |
> |---------------------------|----------------|----------------|----------------|----------------|---------|
> | **Qwen2.5-7B-Instruct**   | 65.10±2.83     | 13.43±3.33     | 3.56±3.97      | 54.79±3.25     | 34.22   |
> | GRPO                      | 90.65±0.90     | 29.98±1.55     | 25.00±1.25     | 70.21±1.76     | 53.96   |
> | DAPO                      | 92.21±0.87     | 31.64±1.42     | 22.80±1.44     | 74.12±1.42     | 55.19   |
> | **PTA-GRPO (Ours)**       | **94.27±0.98** | **34.38±1.25** | **28.47±1.26** | **77.78±1.49** | 58.72   |
> | **Qwen3-8B-Instruct**     | 91.46±2.75     | 68.31±3.60     | 52.34±3.69     | 91.26±3.57     | 75.84   |
> | GRPO                      | **94.04±0.95** | 69.68±1.44     | 55.42±1.39     | 93.51±1.40     | 78.16   |
> | DAPO                      | 93.12±1.01     | 68.36±1.52     | 51.61±1.57     | 92.87±1.52     | 76.49   |
> | **PTA-GRPO (Ours)**       | 94.04±0.98     | **70.51±1.66** | **56.10±1.41** | **94.14±1.80** | 78.70   |
> | **Qwen3-14B-Instruct**    | 91.34±2.53     | 72.22±2.95     | 72.36±3.06     | 95.70±4.11     | 82.91   |
> | GRPO                      | 92.53±1.10     | 73.68±1.65     | 72.27±1.52     | 96.14±1.33     | 83.66   |
> | DAPO                      | 93.03±1.04     | 73.54±1.57     | 72.56±1.62     | 96.58±1.33     | 83.93   |
> | **PTA-GRPO (Ours)**       | **94.38±1.07** | **75.20±1.56** | **73.49±1.57** | **97.56±1.80** | 85.15   |
>
> **Table 4.** Pairwise significance tests between PTA-GRPO and each baseline (Instruct, GRPO, DAPO) with 60K training samples.
> Each row shows: improvement Δ (percentage points), p-value, and significance level.
> `*`, `**`, and `***` denote p < 0.05, p < 0.01, and p < 0.001; `ns` = not significant.
>
> | Model       | Baseline | Task    | Δ (pt) | p / sig      |
> |------------|----------|---------|--------|--------------|
> | Qwen2.5-7B | Instruct | MATH500 | +29.17 | `0.0000***`  |
> | Qwen2.5-7B | Instruct | AIME24  | +20.95 | `0.0000***`  |
> | Qwen2.5-7B | Instruct | AIME25  | +24.90 | `0.0000***`  |
> | Qwen2.5-7B | Instruct | AMC23   | +23.00 | `0.0000***`  |
> | Qwen2.5-7B | GRPO     | MATH500 | +3.62  | `0.0000***`  |
> | Qwen2.5-7B | GRPO     | AIME24  | +4.39  | `0.0000***`  |
> | Qwen2.5-7B | GRPO     | AIME25  | +3.47  | `0.0000***`  |
> | Qwen2.5-7B | GRPO     | AMC23   | +7.57  | `0.0000***`  |
> | Qwen2.5-7B | DAPO     | MATH500 | +2.06  | `0.0000***`  |
> | Qwen2.5-7B | DAPO     | AIME24  | +2.73  | `0.0000***`  |
> | Qwen2.5-7B | DAPO     | AIME25  | +5.66  | `0.0000***`  |
> | Qwen2.5-7B | DAPO     | AMC23   | +3.66  | `0.0000***`  |
> | Qwen3-8B   | Instruct | MATH500 | +2.57  | `0.0000***`  |
> | Qwen3-8B   | Instruct | AIME24  | +2.20  | `0.0000***`  |
> | Qwen3-8B   | Instruct | AIME25  | +3.76  | `0.0000***`  |
> | Qwen3-8B   | Instruct | AMC23   | +2.88  | `0.0000***`  |
> | Qwen3-8B   | GRPO     | MATH500 | -0.01  | `0.9708 ns`  |
> | Qwen3-8B   | GRPO     | AIME24  | +0.83  | `0.0030**`   |
> | Qwen3-8B   | GRPO     | AIME25  | +0.68  | `0.0067**`   |
> | Qwen3-8B   | GRPO     | AMC23   | +0.63  | `0.0279*`    |
> | Qwen3-8B   | DAPO     | MATH500 | +0.91  | `0.0000***`  |
> | Qwen3-8B   | DAPO     | AIME24  | +2.15  | `0.0000***`  |
> | Qwen3-8B   | DAPO     | AIME25  | +4.49  | `0.0000***`  |
> | Qwen3-8B   | DAPO     | AMC23   | +1.27  | `0.0000***`  |
> | Qwen3-14B  | Instruct | MATH500 | +3.03  | `0.0000***`  |
> | Qwen3-14B  | Instruct | AIME24  | +2.98  | `0.0000***`  |
> | Qwen3-14B  | Instruct | AIME25  | +1.12  | `0.0101*`    |
> | Qwen3-14B  | Instruct | AMC23   | +1.86  | `0.0012**`   |
> | Qwen3-14B  | GRPO     | MATH500 | +1.85  | `0.0000***`  |
> | Qwen3-14B  | GRPO     | AIME24  | +1.51  | `0.0000***`  |
> | Qwen3-14B  | GRPO     | AIME25  | +1.22  | `0.0000***`  |
> | Qwen3-14B  | GRPO     | AMC23   | +1.42  | `0.0000***`  |
> | Qwen3-14B  | DAPO     | MATH500 | +1.35  | `0.0000***`  |
> | Qwen3-14B  | DAPO     | AIME24  | +1.66  | `0.0000***`  |
> | Qwen3-14B  | DAPO     | AIME25  | +0.93  | `0.0013**`   |
> | Qwen3-14B  | DAPO     | AMC23   | +0.98  | `0.0007***`  |

---

> > ### Comment · Reviewer_rX9v · 2025-11-25
> >
> > I appreciate the effort to include the standard deviation and p-value, but you can see that the gains diminish as the models get larger, and the error ranges between different methods start to overlap. It’s difficult to see meaningful practical value for real use cases with large models.

---

> > > ### Author Response · Authors · 2025-11-25
> > >
> > > Dear Reviewer rX9v
> > >
> > > I truly appreciate your response !
> > >
> > > I will solve the remaining concerns for you about the performance on larger LLM. Although the performance gains become smaller as the model size increases and the error ranges across methods begin to overlap, we conducted **t-test significance analyses** based on multiple runs (Table 2). The results show that even for larger models, the improvements remain **statistically significant**, indicating that they are not caused by random variation.
> > >
> > > It is important to emphasize that the smaller gains observed in large models mainly result from the **limited size of the RL training data (only 14K samples)**. This is a **data-scale limitation, not a limitation of the method itself**. When we expand the RL dataset to **60K**, the improvements become substantially larger. As demonstrated in Table 3 and Table 4, with a larger-scale RL dataset, our method achieves **much more pronounced gains**, along with improved stability and consistency across datasets.
> > >
> > > I hope my explanation has addressed your concerns, and I look forward to continuing our communication.
> > >
> > >
> > > Best Regard
> > >
> > > All authors of Submission 19425

---

> ### Author Response · Authors · 2025-11-23
>
> ## **Weakness 2: Unjustified Computational Cost**
>
> We believe this concern arises from a misunderstanding of our sampling scheme.
> In our implementation, the number of forward passes per question for PTA-GRPO is **the same** as for the GRPO and DAPO baselines.
>
> Specifically, for each question, PTA-GRPO samples $m$ analytic plans and, for each plan, $z$ CoTs, which yields $m \times z$ rollout trajectories in total. Meanwhile, GRPO and DAPO in our experiments are also trained with exactly $m \times z$ rollout trajectories per question, using the same sampling budget.
>
> In other words, the total number of model forward passes is matched across all methods; PTA-GRPO only restructures these $m \times n$ trajectories into plan-centric groups and does **not** introduce an extra $3\times$ computational overhead compared to GRPO. Therefore, the performance gains of PTA-GRPO do *not* stem from using more samples, but from the proposed planning-and-training mechanism itself.
>
> **From a practical perspective, during training on 8 H200 GPUs, GRPO requires 20.7 hours, DAPO requires 20.3 hours, and PTA-GRPO requires 21.2 hours. The time differences among the three are negligible.**
>
> We would like to stress that this equal-compute setting was *already specified* in the original submission, both in the caption of Fig. 2 and in the description of the baselines. However, we acknowledge that this clarification may not have been sufficiently prominent and could be easily overlooked. In the revised version, we will further highlight this point and explicitly state that PTA-GRPO, GRPO, and DAPO are compared under the same compute budget and number of forward passes.

---

> > ### Comment · Reviewer_rX9v · 2025-11-25
> >
> > I understand that you can increase the cost for GRPO and DAPO by using the same number of rollouts, but they should already perform well with $n$ rollouts. Based on existing literature, increasing the number of rollouts does not yield significant improvement. Should you include another baseline for GRPO and DAPO using $n$ rollouts instead?

---

> > > ### Author Response · Authors · 2025-11-25
> > >
> > > Dear Reviewer rX9v
> > >
> > > Thank you very much for your comment. We believe there may be a small misunderstanding. **While it is sometimes assumed that 3–4 rollouts are sufficient for GRPO, multiple studies [1, 2] actually show that this is often not the case. In fact, [2] demonstrates clear performance improvements all the way up to 64 rollouts, with no saturation before that point. In fact, in the original GRPO report [3], they also chose a rollout size of 64 rather than 3 to 4.**
> > >
> > > Given this evidence, our setting is a reasonable and well-supported choice for a fair GRPO baseline, and it avoids underestimating GRPO’s potential.
> > >
> > > Regarding your point, “Based on existing literature, increasing the number of rollouts does not yield significant improvement,” we were unable to find research supporting this claim. We kindly ask whether you could share the papers or references that present this viewpoint, so that we can engage in a deeper and more informed discussion.
> > >
> > > We sincerely appreciate your feedback, and we hope this clarification resolves the concern. If you have any further concerns, we would be happy to discuss them with you.
> > >
> > > Best Regard
> > >
> > > Your Friends
> > >
> > > All authors of Submission 19425
> > >
> > > [1] Tan, Z., Geng, H., Zhang, M., Yu, X., Wan, G., Zhou, Y., ... & Bai, L. (2025). Scaling Behaviors of LLM Reinforcement Learning Post-Training: An Empirical Study in Mathematical Reasoning. arXiv preprint arXiv:2509.25300.
> > >
> > > [2] Xu, Y. E., Savani, Y., Fang, F., & Kolter, J. Z. (2025). Not all rollouts are useful: Down-sampling rollouts in llm reinforcement learning. arXiv preprint arXiv:2504.13818.
> > >
> > > [3] Shao, Z., Wang, P., Zhu, Q., Xu, R., Song, J., Bi, X., ... & Guo, D. (2024). Deepseekmath: Pushing the limits of mathematical reasoning in open language models. arXiv preprint arXiv:2402.03300.

---

> > > ### Author Response · Authors · 2025-11-25
> > >
> > > Dear Reviewer rX9v
> > >
> > > Thank you very much for your comment. **We believe there may be a small misunderstanding. While it is sometimes assumed that 3–4 rollouts are sufficient for GRPO, multiple studies [1, 2] actually show that this is often not the case. In fact, [2] demonstrates clear performance improvements all the way up to 64 rollouts, with no saturation before that point. In fact, in the original GRPO report [3], they also chose a rollout size of 64 rather than 3 to 4.**
> > >
> > > Given this evidence, our setting is a reasonable and well-supported choice for a fair GRPO baseline, and it avoids underestimating GRPO’s potential.
> > >
> > > Regarding your point, “Based on existing literature, increasing the number of rollouts does not yield significant improvement,” we were unable to find research supporting this claim. We kindly ask whether you could share the papers or references that present this viewpoint, so that we can engage in a deeper and more informed discussion.
> > >
> > > We sincerely appreciate your feedback, and we hope this clarification resolves the concern. If you have any further concerns, we would be happy to discuss them with you.
> > >
> > > Best Regard
> > >
> > > Your Friends
> > >
> > > All authors of Submission 19425
> > >
> > > [1] Tan, Z., Geng, H., Zhang, M., Yu, X., Wan, G., Zhou, Y., ... & Bai, L. (2025). Scaling Behaviors of LLM Reinforcement Learning Post-Training: An Empirical Study in Mathematical Reasoning. arXiv preprint arXiv:2509.25300.
> > >
> > > [2] Xu, Y. E., Savani, Y., Fang, F., & Kolter, J. Z. (2025). Not all rollouts are useful: Down-sampling rollouts in llm reinforcement learning. arXiv preprint arXiv:2504.13818.
> > >
> > > [3] Shao, Z., Wang, P., Zhu, Q., Xu, R., Song, J., Bi, X., ... & Guo, D. (2024). Deepseekmath: Pushing the limits of mathematical reasoning in open language models. arXiv preprint arXiv:2402.03300.

---

> > > ### Author Response · Authors · 2025-11-27
> > >
> > > Dear Reviewer rX9v
> > >
> > >
> > > As per your requirements, we conducted experiments using analytic plan = 2, CoT = 2 for each plan, and a group rollout size of 4, and compared them with GRPO using a rollout size of 4. The results are shown in the table 1 below.
> > >
> > > **Table 1: Performance Comparison between GRPO and PTA-GRPO with group rollout size 4, where Qwen2.5-7B-Instruct is considered as base model.**
> > >
> > > | Method   | MATH500 | AIME24 | AIME25 | AMC23 |
> > > |----------|---------|--------|--------|--------|
> > > | GRPO     | 80.6    | 23.7   | 20.4   | 61.9   |
> > > | PTA-GRPO | **82.2** | **26.9** | **22.7** | **65.4** |
> > >
> > > **The results lead to two main conclusions. First, PTA-GRPO achieves consistent improvements over GRPO even when the rollout size is as small as 4. Second, a rollout number of 4 is still insufficient compared to 12, which is consistent with the observations reported in previous works [1, 2, 3], indicating that our experimental setting is reasonable.**
> > >
> > > I hope this clarifies all of your concerns. Please let me know if you have any further questions—I'd be very happy to discuss them in more depth.
> > >
> > > Best regards,
> > >
> > > Your friends
> > >
> > > All authors of Submission 19425
> > >
> > >
> > > [1] Tan, Z., Geng, H., Zhang, M., Yu, X., Wan, G., Zhou, Y., ... & Bai, L. (2025). Scaling Behaviors of LLM Reinforcement Learning Post-Training: An Empirical Study in Mathematical Reasoning. arXiv preprint arXiv:2509.25300.
> > >
> > > [2] Xu, Y. E., Savani, Y., Fang, F., & Kolter, J. Z. (2025). Not all rollouts are useful: Down-sampling rollouts in llm reinforcement learning. arXiv preprint arXiv:2504.13818.
> > >
> > > [3] Shao, Z., Wang, P., Zhu, Q., Xu, R., Song, J., Bi, X., ... & Guo, D. (2024). Deepseekmath: Pushing the limits of mathematical reasoning in open language models. arXiv preprint arXiv:2402.03300.

---

> ### Author Response · Authors · 2025-11-23
>
> ## **Weakness 3: Lack of Novelty and weak theorcial analysis**
>
> Regarding the reviewer’s concerns about the scope and novelty of our theoretical contributions, we fully understand and appreciate the feedback. In the revision, we will clearly separate our contributions into three parts:
>
> 1. The conceptual innovation of the **plan-then-action architecture**,
> 2. The **RL algorithmic contribution** enabled by multi-sample plan evaluation, and
> 3. The **theoretical interpretation** that explains why plan quality influences reasoning performance through conditional mutual information.
>
> ### **1 Innovation in the Plan-then-Action Framework:**
>
> We want to highlight **our notion of plan–action is fundamentally different from CoT, ReAct, ToT, and least-to-most prompting**. In detail:
>
> - **CoT** produces a reasoning trace, a token-level step-by-step derivation, which captures low-level details rather than a high-level analytic structure.
> - **ReAct** interleaves actions and observations in a sequential decision process, focusing on tool usage and interaction rather than conceptual decomposition; its “plan” is operational, not analytical.
> - **ToT** constructs a search tree over multiple reasoning branches, and its notion of planning is tied to inference-time branching and selection; it does not provide a stable, reusable analytic scaffold and cannot be internalized as a single coherent plan.
> - **Least-to-most prompting** decomposes a problem via prompting instructions, but the decomposition itself is not an explicit analytic representation generated by the model—rather, it is externally imposed and tightly coupled to a decomposition-then-solving protocol.
>
> **In our work, we propose an innovative framework where a single compact response simultaneously conveys high-level ideas (the plan) and fine-grained chain-of-thought reasoning. By first generating an explicit plan that directly guides the subsequent CoT, the model’s reasoning becomes more structured, efficient, and compact.**
>
> **Crucially, previous mentioned paradigms do not provide a well-defined model-generated analytic plan that can be directly optimized by using reinforcement learning, making it extremely difficult to systematically strengthen their planning quality or execution reliability through RL.** In contrast, our plan–action formulation produces a compact response in which the model first generates an abstract analytic idea and then executes it through a downstream CoT process. Concretely, our outputs follow a fixed “plan + CoT” format (`<plan>...</plan>` followed by `<think>...</think>`), presented within a single response. This structure is particularly suitable for RLVR optimization, since both the plan and the subsequent CoT can be evaluated solely through their contribution to the final answer. The plan $t$ functions as an analytic scaffold, providing a concise, problem-level abstraction that guides and elicits the subsequent CoT reasoning at a fine-grained level.

---

> ### Author Response · Authors · 2025-11-23
> **Weakness 3 (continued) (1)**
>
> ## **Weakness 3 (continued) (1):**
>
> ### **2 RL contribution is not a straightforward “GRPO + auxiliary reward” extension**
>
> We appreciate the reviewer's observation that our method builds on GRPO. However, we respectfully disagree that PTA-GRPO is merely a straightforward “GRPO + auxiliary reward” extension.
> **PTA-GRPO introduces innovations to the RL rollout architecture: it incorporates a hierarchical reinforcement learning process into the original GRPO framework, while also compensating for RLVR’s lack of supervision over intermediate reasoning steps.**
>
> Existing RLVR approaches, such as GRPO and DAPO, rely almost exclusively on the correctness of the final answer when computing rewards, which means that the intermediate reasoning process cannot be directly constrained. Concretely, these methods generate a group of CoT responses $\lbrace c\_i \rbrace\_{i=1}^N$ and assign a corresponding reward set $\lbrace r\_i\rbrace \_{i=1}^N$ based solely on the correctness of their final outputs. As a result, the reasoning trajectories within each CoT $c\_i$ receive only indirect supervision rather than being explicitly optimized.
>
> Unlike traditional approaches, PTA-GRPO adopts a structured reasoning path composed of a plan and a CoT, denoted as $(t, c)$. For each plan $t$, the model samples multiple CoT trajectories $\{c_j\}_{j=1}^z$, where $c_j \sim \pi(\cdot \mid q, t)$. The correctness probabilities of these CoT trajectories are used both to evaluate the quality of the plan and to provide reward signals for the whole response.
>
> Consequently, within PTA-GRPO, PSG-RL decomposes the training signal into two components:
> (i) an outcome reward $r_{\text{outcome}}$, defined as
> $$
> r\_{\text{outcome}}(c_j) = \mathbb{I}[\hat{y}\_j = y],
> $$
> and (ii) an analytic plan reward $r_{\text{analytic}}$, defined as
> $$
> r\_{\text{analytic}}(t)
> = \operatorname{Softmax}\left(\frac{1}{z} \sum_{j=1}^z r_{\text{outcome}}(c_j)\right).
> $$
>
> Moreover, unlike prior RLVR methods that rely solely on the outcome reward $r_{\text{outcome}}$, our framework introduces an additional analytic plan reward $r_{\text{analytic}}$. This reward assigns higher utility to plans $t$ whose induced trajectories achieve higher empirical success rates, indicating that the corresponding bundle of trajectories is of higher overall quality.
>
> Therefore, the analytic plan reward $r_{\text{analytic}}$ provides a soft evaluation and constraint on the intermediate reasoning process: on the one hand, it encourages the model to optimize toward reasoning paths that are more likely to succeed, while on the other hand it avoids completely rejecting the entire intermediate reasoning process solely because the final answer is incorrect.
>
> By explicitly optimizing over plans’ reward in this way, PTA-GRPO imposes a much stronger constraint on the intermediate reasoning process, which existing RLVR methods do not explicitly regulate.

---

> ### Author Response · Authors · 2025-11-23
> **Weakness 3 (continued) (2):**
>
> ## Weakness 3 (continued) (2):
>
>
> ### **Weak contribution:**
>
> We would like to clarify that our theoretical analysis is not “generic and tautological.” The result does not merely restate that “better plans lead to lower error,” but instead provides a **concrete and interpretable information-theoretic mechanism**: when a plan $t$ is of higher quality, the conditional prediction distribution $\pi(\hat{y} \mid q, t)$ becomes more aligned with the true answer distribution, making the prediction $\hat{y}$ more informative about the correct label $y$, i.e., **increasing the conditional mutual information $I(\hat{y}; y \mid q, t)$**.
>
> According to Theorem~3.1, a larger mutual information directly tightens the upper bound on the error probability. Therefore, the theorem not only establishes a meaningful and quantifiable link between “improved plan quality” and “reduced reasoning error,” but also shows that enhancing plan quality through analytic rewards increases the probability of obtaining the correct answer $\Pr(y = \hat{y} \mid t, q)$. This, in turn, strengthens the relationship between the predicted output $\hat{y}$ and the ground-truth $y$, thereby increasing their mutual information $I(y;\hat{y} \mid t,q)$, where
> $$
> I(y;\hat{y} \mid t,q) = H(y\mid q,t) - H(y \mid \hat{y},q,t).
> $$
>
> More concretely, for a fixed data distribution, the conditional entropy $H(y \mid q, t)$ is determined by the question and does not depend on the predicted answer $\hat{y}$. In our framework, the analytic plan reward $r_{\text{analytic}}$ is constructed to maximize a Softmax-transformed estimate of the success probability $\Pr(\hat{y} = y \mid q, t)$. As this probability increases, the predicted answer $\hat{y}$ becomes more informative about the ground-truth label $y$, and the remaining uncertainty $H(y \mid \hat{y}, q, t)$ decreases. Equivalently, the conditional mutual information
> $$
> I(\hat{y}; y \mid q, t) = H(y \mid q, t) - H(y \mid \hat{y}, q, t)
> $$
> increases, which directly tightens the error bound in our information-theoretic analysis.
>
> In addition, we also appreciate the suggestion of the reviewer on the optimality of the analytic reward. We agree that clarifying this connection would strengthen the paper. In response, we have added a new **Proposition** in the revision, showing that our analytic plan reward is a monotonic surrogate for the conditional success probability $\Pr(\hat{y} = y \mid q, t)$ and therefore directly encourages increases in $I(\hat{y}; y \mid q, t)$. This proposition further supports the theoretical grounding of our reward design and addresses the reviewer’s concern.
>
> **Proposition.** Let $t_1$ and $t_2$ be two analytic plans, and let $r_{\mathrm{analytic}}(t_1)$ and $r_{\mathrm{analytic}}(t_2)$ denote their corresponding analytic rewards. Let $\hat{y}_1$ and $\hat{y}_2$ be the answers induced by executing $t_1$ and $t_2$, respectively. If
> $$
> r\_{\mathrm{analytic}}(t\_1) \ge r\_{\mathrm{analytic}}(t\_2),
> $$
> it holds that
> $$
> H(y \mid \hat{y}\_1,t\_1,q) \leq H(y \mid \hat{y}\_2,t\_2,q).
> $$
>
> **The proof can be seen in Appendix B.7.**  Specifically, a higher analytic reward $r\_{\text{analytic}}$ implies a lower evaluation score $H(y \mid \hat{y}, t, q)$, which is monotonically aligned with the likelihood of producing the correct answer. Because $\Pr(\hat{y} = y \mid q, t)$ is a monotonic decreasing function of the conditional entropy $H(y \mid \hat{y}, q, t)$, we have
> $
> \max r\_{\text{analytic}} \ \Longleftrightarrow\ \min H(y \mid \hat{y}, q, t).
> $
> Moreover, since $I(y;\hat{y}\mid q,t) = H(y\mid q,t) - H(y\mid \hat{y},q,t)$ and $H(y\mid q,t)$ is independent of $t$, minimizing the conditional entropy is also equivalent to maximizing the conditional mutual information. Hence, we obtain the equivalence
> $
> \max r\_{\mathrm{analytic}}(t)\ \Longleftrightarrow\
> \min H(y\mid \hat{y},q,t)\  \Longleftrightarrow\ \max I(y;\hat{y}\mid q,t).
> $
> Therefore, we have shown that our analytic reward increases the conditional mutual information between the analytic plan and the correct answer, thereby reducing the answer error rate.
>
> Moreover, to further verify the correctness of our theoretical analysis, we empirically investigate the relationship between the analytic plan reward $r\_{\text{analytic}}$ and the mutual information between the model's predicted answer $\hat{y}$ and the ground-truth answer $y$ in Section 3.3. The experimental results show that $r\_{\text{analytic}}$ is significantly positively correlated with $I(\hat{y}, y)$, indicating that $r\_{\text{analytic}}$ indeed captures the degree of information coupling between the predictions and the true labels, thereby demonstrating that our theory is effective in practice.
>
> In addition, we also show the empirical analysis in Section 3.3 to support and validate our theoretical findings.

---

> ### Author Response · Authors · 2025-11-23
> **Weakness 3 (continued) (3):**
>
> ## **Weakness 3 (continued) (3):**
>
> ### Proof of proposition
>
> **Proof.**
> Let $p\_i = \Pr(\hat{y}\_i = y \mid t\_i, q)$ denote the probability that the final answer generated under plan $t\_i$ is correct. The analytic reward $r\_{\mathrm{analytic}}(t\_i)$ is a monotonic function of the empirical estimate of $p\_i$. Therefore, the condition $r\_{\mathrm{analytic}}(t\_1) \ge r\_{\mathrm{analytic}}(t\_2)$ implies:
> $$
> p\_1 \ge p\_2. \tag{1}
> $$
>
> The conditional entropy $H(y \mid \hat{y}\_i, t\_i, q)$ measures the remaining uncertainty about the true answer $y$ after observing the predicted answer $\hat{y}_i$ generated under plan $t\_i$. We decompose this entropy by conditioning on whether $\hat{y}\_i$ is correct:
> $$
> H(y \mid \hat{y}\_i, t\_i, q) =  \Pr(\hat{y}\_i = y \mid t\_i, q) \cdot H(y \mid \hat{y}\_i=y, t\_i, q) + \Pr(\hat{y}\_i \neq y \mid t\_i, q) \cdot \mathbb{E}[H(y \mid \hat{y}\_i, t\_i, q) \mid \hat{y}\_i \neq y].
> $$
>
> If the predicted answer $\hat{y}\_i$ is correct (i.e., the event $\hat{y}\_i = y$ occurs), then the posterior distribution $\Pr(y \mid \hat{y}\_i, t\_i, q)$ collapses to a point mass on the value $\hat{y}\_i$, resulting in zero conditional entropy:
> $
> H(y \mid \hat{y}\_i=y, t\_i, q) = 0.
> $
>
> Let $
> C\_i = \mathbb{E}[H(y \mid \hat{y}\_i, t_i, q) \mid \hat{y}\_i \neq y]$
> denote the expected conditional entropy when the predicted answer is wrong. Substituting into the equation above yields:
> $$
> H(y \mid \hat{y}\_i, t\_i, q) = p\_i \cdot 0 + (1 - p\_i) \cdot C\_i = (1 - p\_i) C\_i. \tag{2}
> $$
>
> We now compare the entropies for the two plans:
> $$
> H(y \mid \hat{y}\_1, t\_1, q) = (1 - p\_1) C\_1, \quad
> H(y \mid \hat{y}\_2, t\_2, q) = (1 - p\_2) C\_2.
> $$
>
> From (1), we have $1 - p\_1 \le 1 - p\_2$.
>
> We now introduce the core assumption: a plan with a higher analytic reward provides more informative guidance, leading to a posterior distribution over \(y\) that is more concentrated even when the predicted answer is incorrect. Formally, this means:
> $$
> C\_1 \le C\_2. \tag{3}
> $$
>
> This is reasonable because a high-quality plan constrains the reasoning path more effectively, reducing the set of plausible wrong answers and resulting in lower uncertainty upon observing an incorrect prediction.
>
> Since $(1 - p_1) \le (1 - p_2)\) and \(C_1 \le C_2$, and all terms are non-negative, it follows that:
> $$
> (1 - p\_1) C\_1 \le (1 - p\_2) C\_2.
> $$
>
> Therefore, by (2):
> $$
> H(y \mid \hat{y}\_1, t\_1, q) \le H(y \mid \hat{y}\_2, t\_2, q),
> $$
> which completes the proof. $\square$

---

> ### Author Response · Authors · 2025-11-23
>
> ## **Weakness 4: Narrow domains and lack of baselines**
>
> ### **Lack of baselines** We thank the reviewer for pointing this out and we agree that a clearer comparison to planning and reflection approaches is important. We compare our method with these approaches, and the results of this comparison are shown in Table 4. Experimental results show that PTA-GRPO achieves significantly superior performance across all evaluation datasets, far surpassing the other baseline methods.
>
> **Table 4.** Comparison of different baselines, where Qwen2.5-7B-Instruct is taken as the base model.
> | Method      | MATH500 | AIME24 | AIME25 | AMC23 | Average |
> |------------|---------|--------|--------|-------|---------|
> | ToT        | 72.93   | 12.74  | 9.55   | 58.50 | 38.43   |
> | ReAct      | 70.55   | 11.55  | 9.75   | 60.22 | 38.02   |
> | Gandhi et al | 80.52 | 24.95  | 20.03  | 64.29 | 47.45   |
> | PTA-GRPO     | **85.57** | **30.26** | **25.97** | **70.24** | **53.01** |
>
>
> ### **Narrow experiment**
>
> Thank you for raising this insightful concern. We agree that the original experiments focused primarily on mathematical reasoning. In Table 5, we have extended PTA-GRPO to the multimodal setting and conducted additional evaluations on both General-Benchmark and Science Benchmark, using Qwen2.5-7B-VL as the base model.
>
> For a fair comparison, our SFT and RL datasets follow the same setup as [1]. For the general benchmarks, we evaluate on MMMU-Pro [2], MMMU [3], and EMMA [4], while for the science benchmark, we adopt MMK12 [5].
>
> These new results further demonstrate the generality and robustness of our approach beyond the math domain.
>
> **Table 5.** Comparison between PTA-GRPO and other approaches on General-Benchmark and Science Benchmark, using Qwen2.5-7B-VL as the base model.
>
> | Method        | MMMU-Pro | MMMU  | EMMA | Phys  | Chem  | Bio   |
> |--------------|----------|-------|------|-------|-------|-------|
> | **Base**     | 36.9     | 54.3  | 21.5 | 45.4  | 56.4  | 54.0  |
> | MM-Eurek     | 37.6     | 55.2  | 23.5 | 45.4  | 56.4  | 54.0  |
> | SRPO         | 42.3     | 57.1  | 29.6 | 56.2  | 65.2  | 65.2  |
> | PTA-GRPO     | **44.7** | **59.0** | **31.9** | **58.5** | **68.7** | **66.8** |
>
>
> [1] Wan, Z., Dou, Z., Liu, C., Zhang, Y., Cui, D., Zhao, Q., ... & Yan, S. (2025). Srpo: Enhancing multimodal llm reasoning via reflection-aware reinforcement learning. NeurIPS 2025.
>
> [2] Yue, X., Zheng, T., Ni, Y., Wang, Y., Zhang, K., Tong, S., ... & Neubig, G. (2025, July). Mmmu-pro: A more robust multi-discipline multimodal understanding benchmark. In Proceedings of the 63rd Annual Meeting of the Association for Computational Linguistics (Volume 1: Long Papers) (pp. 15134-15186).
>
> [3] Yue, X., Ni, Y., Zhang, K., Zheng, T., Liu, R., Zhang, G., ... & Chen, W. (2024). Mmmu: A massive multi-discipline multimodal understanding and reasoning benchmark for expert agi. In Proceedings of the IEEE/CVF Conference on Computer Vision and Pattern Recognition (pp. 9556-9567).
>
> [4] Standley, T., Gao, R., Chen, D., Wu, J., & Savarese, S. (2023, January). An extensible multi-modal multi-task object dataset with materials. In International Conference on Learning Representations.
>
> [5] Meng, F., Du, L., Liu, Z., Zhou, Z., Lu, Q., Fu, D., ... & Shao, W. (2025). Mm-eureka: Exploring the frontiers of multimodal reasoning with rule-based reinforcement learning. arXiv preprint arXiv:2503.07365.
>
> **We once again thank the reviewer for their constructive comments, which have helped us improve the quality of the paper. We hope that the revised version addresses your concerns. If you have any further questions or suggestions, please feel free to let us know.**
>
> Best wishes and regards,
>
> All authors of Submission 19425

---

> ### Author Response · Authors · 2025-11-25
>
> Dear Reviewer rX9v,
>
> We would like to express our sincere gratitude for your constructive comments and suggestions on our manuscript. Your feedback has been very helpful and has significantly improved the overall quality and clarity of the paper.
>
> During the rebuttal process, we carefully addressed your concerns and revised the manuscript accordingly. We hope that our responses and the changes made in the revised version have resolved the issues you raised.
>
> If you have any remaining questions, concerns, or additional suggestions, we would be very glad to further discuss them and provide any clarifications that may be helpful.
>
> Thank you again for your time and effort in reviewing our work.
>
> Best regards,
>
> All authors of Submission 19425

---

### Official Review · Reviewer_o4Ko · 2025-11-01

**Soundness:** 3
**Presentation:** 4
**Contribution:** 3
**Rating:** 8
**Confidence:** 3

**Summary:**

The paper proposes Plan-Then-Action enhancement to GRPO (PTA-GRPO), a two stage training framework for enhanced LLM reasoning tasks. It adds an additional "plan" part before the standard "think" part in deep reasoning. It uses an SFT stage for the "plan", and a RL stage using GRPO for the "plan" and "think" jointly. Experiments show strong performance over GRPO and similar methods.

**Strengths:**

+ The method is intuitive, focusing on a major disadvantage in LLM deep reasoning -- lack of global guidance, which causes many additional token consumption of redundancy and off-topic reasoning.
+ The two-stage training ensures a smooth learning of the additional "plan" part. The second stage features the newly proposed Planning Structure-Guided variant of GRPO. The technical method is novel.
+ The additional analytical plan reward fits the new RL training.
+ The framework, dataset, and algorithms are clearly explained in the paper.
+ Experiments show consistent performance better than GRPO and other similar methods across popular mathematical reasoning methods across different LLM architecture and sizes.

**Weaknesses:**

+ Theoretically, the standard GRPO can trace back and refine its action steps if it makes a mistake. However, the proposed PTA-GRPO has a fixed plan, which may be not accurate before execution but lacks a mechanism to correct itself ***during inference***
+ It lacks possible experiments with larger models, which is known better at reasoning, while small models are generally considered not very suitable for GRPO fine-tuning and reasoning. Though such training requires extensive resources, I still recommend include some as baselines for better demonstration.

**Questions:**

+ If possible, I suggest include prompting and SFT-based (including stage 1 results) methods in Table 1, further confirming the importance of the high level plan. It's also good to include baseline numbers from top model (e.g. GPT-5) experiments / technical reports.

---

> ### Author Response · Authors · 2025-11-22
>
> Dear Reviewer o4Ko
>
> We sincerely appreciate your support for this paper and your recognition of our work. We hope that the upcoming rebuttal will clear up your concerns.
>
> ## **Weakness 1: the plan’s misleading guidance**
>
> We thank the reviewer for this constructive comment. While PTA-GRPO indeed starts from a fixed high-level plan, it is not strictly bound to this initial plan during inference. Following [1], we explicitly adjust the prompts used in PTA-GRPO to encourage active self-reflection and plan revision during the action phase (see the prompts in the Appendix). Concretely, if the initial plan is suboptimal or partially incorrect, the LLM is guided to detect inconsistencies and correct its subsequent actions accordingly. We have added illustrative examples of such behavior in the rebuttal and Appendix B.4 to make this clearer. We once again thank the reviewer for the suggestion, and we will clarify this further in the revision.
>
> ## **Weakness 2:  Performance on larger model**
>
> We appreciate the reviewer’s insightful suggestion. We fully agree that larger models, which are generally stronger at reasoning, would be valuable baselines for further demonstrating the effectiveness of our method. In our preliminary attempts with stronger models (e.g., Qwen3-14B's results in main paper), we observed that using only a relatively small amount of training data leads to very limited performance gains, suggesting that substantial improvements for such large models would require both significantly larger-scale datasets and much higher computational cost. Unfortunately, this exceeds our current resource budget.
> However, based on the current results, it still plays an important role in improving small LLMs and has a significant impact.
> In the future, we will therefore leave a more systematic investigation on larger models and large-scale training as an important direction for future work.
>
> ## **Question 1**
>
> We appreciate the reviewer’s helpful suggestion. In the current version of the paper, Table 3 already reports the Stage-1 (SFT-based) results, which can be viewed as a baseline that helps illustrate the contribution of the high-level plan. We agree that presenting prompting and SFT-based methods together with PTA-GRPO in the main results table would make the comparison clearer and further highlight the importance of the plan. In the final revision, we will (i) reorganize the tables so that prompting and Stage-1 (SFT-based) results are included alongside our GRPO-based methods, and (ii) add baseline numbers from top models (e.g., GPT-5) based on publicly available technical reports, to provide a more comprehensive demonstration.
>
> **We are once again deeply grateful for your support of this work. Should you have any questions, please feel free to tell us.**
>
> Best wishes and regards,
>
> All authors of Submission 19425
>
>
> [1] Gandhi, K., Chakravarthy, A., Singh, A., Lile, N., & Goodman, N. D. (2025). Cognitive behaviors that enable self-improving reasoners, or, four habits of highly effective stars. arXiv preprint arXiv:2503.01307.

---

> ### Author Response · Authors · 2025-11-22
> **Example of an LLM fixing its plan through self-reflection**
>
> **Question:**
> Six points \( A, B, C, D, E, \) and \( F \) lie in a straight line in that order. Suppose that \( G \) is a point not on the line and that \( AC = 26 \), \( BD = 22 \), \( CE = 31 \), \( DF = 33 \), \( AF = 73 \), \( CG = 40 \), and \( DG = 30 \). Find the area of \( \triangle BGE \).
>
> ---
>
> ### \<plan>
>
> 1. Assign coordinates to points A through F on the x-axis with A at 0.
> 2. Use the given segment lengths (AC, BD, CE, DF, AF) to set up equations and solve for all individual segment lengths \(AB, BC, CD, DE, EF\).
> 3. Determine the coordinates of B, C, D, E, F.
> 4. Use distances \(CG\) and \(DG\) to locate point \(G\) by solving a system of equations from the distance formula.
> 5. Calculate area of triangle \(BGE\) using the shoelace formula with coordinates \(B, G, E\).
>
> > **Note:** Wrong plan.
>
> ---
>
> ### \<think>
>
> I notice the plan is incorrect in its approach to finding point \(G\). The plan says to use distances \(CG\) and \(DG\) to locate \(G\), but this will only give us the x-coordinate of \(G\), not both coordinates. Let me revise this.
>
> > **Note:** Revise it by LLM self-reflection
>
> Looking more carefully at the problem, I need to find the area of triangle \(BGE\). The points \(B, G,\) and \(E\) form a triangle where:
>
> - \(B\) and \(E\) are on the x-axis
> - \(G\) is somewhere in the plane
>
> The area of triangle \(BGE\) can be calculated as
> \[
> \frac{1}{2} \times |BE| \times \text{height from } G \text{ to line } BE.
> \]
> Since \(BE\) is horizontal (both \(B\) and \(E\) on the x-axis), the height is simply \(|y\text{-coordinate of }G|\).
>
> So the revised approach should be:
>
> 1. Find coordinates of \(B\) and \(E\).
> 2. Find the y-coordinate of \(G\).
> 3. Use \(\text{Area} = \frac{1}{2} \times |BE| \times |y_G|\).
>
> ---
>
> First, let's find all segment lengths as in the original plan. Let
> \[
> AB = a,\quad BC = b,\quad CD = c,\quad DE = d,\quad EF = e.
> \]
>
> From the given data:
>
> - From \(AC = 26\):
>   \[
>   a + b = 26
>   \]
> - From \(BD = 22\):
>   \[
>   b + c = 22
>   \]
> - From \(CE = 31\):
>   \[
>   c + d = 31
>   \]
> - From \(DF = 33\):
>   \[
>   d + e = 33
>   \]
> - From \(AF = 73\):
>   \[
>   a + b + c + d + e = 73
>   \]
>
> From \(a + b = 26\) and \(a + b + c + d + e = 73\), we get:
> \[
> c + d + e = 47
> \]
>
> From \(c + d = 31\) and \(c + d + e = 47\), we get:
> \[
> e = 16
> \]
>
> From \(d + e = 33\) and \(e = 16\), we get:
> \[
> d = 17
> \]
>
> From \(c + d = 31\) and \(d = 17\), we get:
> \[
> c = 14
> \]
>
> From \(b + c = 22\) and \(c = 14\), we get:
> \[
> b = 8
> \]
>
> From \(a + b = 26\) and \(b = 8\), we get:
> \[
> a = 18
> \]
>
> So we have:
>
> - \(AB = 18\)
> - \(BC = 8\)
> - \(CD = 14\)
> - \(DE = 17\)
> - \(EF = 16\)
>
> Place \(A\) at \((0,0)\). Then:
>
> - \(B = (18, 0)\)
> - \(C = (26, 0)\)
> - \(D = (40, 0)\)
> - \(E = (57, 0)\)
> - \(F = (73, 0)\)
>
> ---
>
> Now, to find the coordinates of \(G\):
>
> We know \(CG = 40\) and \(DG = 30\). Let \(G = (x,y)\).
>
> From the distance formula:
>
> \[
> CG^2 = (x - 26)^2 + y^2 = 1600
> \]
> \[
> DG^2 = (x - 40)^2 + y^2 = 900
> \]
>
> Subtract the second equation from the first:
>
> \[
> (x - 26)^2 - (x - 40)^2 = 700
> \]
>
> Expand:
>
> \[
> (x^2 - 52x + 676) - (x^2 - 80x + 1600) = 700
> \]
>
> \[
> 28x - 924 = 700
> \]
>
> \[
> 28x = 1624 \Rightarrow x = 58
> \]
>
> Now plug back into \((x - 26)^2 + y^2 = 1600\):
>
> \[
> (58 - 26)^2 + y^2 = 1600
> \]
>
> \[
> 32^2 + y^2 = 1600
> \]
>
> \[
> 1024 + y^2 = 1600
> \]
>
> \[
> y^2 = 576 \Rightarrow y = 24
> \]
>
> (Take the positive value; the negative would give a congruent triangle with the same area.)
>
> So:
> \[
> G = (58, 24)
> \]
>
> ---
>
> Now consider triangle \(BGE\):
>
> - \(B = (18, 0)\)
> - \(E = (57, 0)\)
> - \(G = (58, 24)\)
>
> Compute the base:
>
> \[
> BE = 57 - 18 = 39
> \]
>
> The height is the y-coordinate of \(G\), which is 24.
>
> So the area is:
>
> \[
> \text{Area} = \frac{1}{2} \times 39 \times 24 = 468
> \]
>
> ---
>
> **Answer:**
>
> \[
> \boxed{468}
> \]

---

### Author Response · Authors · 2025-11-25
**Summary of Our Rebuttal**

Dear all reviewers,

We sincerely appreciate the reviewers’ valuable and constructive comments, which have greatly helped us improve the clarity and quality of the manuscript. In this rebuttal, we carefully address the main concerns raised by all reviewers. For Reviewer o4Ko’s concern about how an incorrect plan may lead to subsequent erroneous reasoning, we provide further analysis and clarification. For Reviewers rX9v and 8TPm, who questioned the computational cost and the generalization ability of PTA-GTPO across different tasks, we add extensive experimental results to support our claims. Regarding Reviewer rX9v’s concerns about the stability of our method, we include additional experiments showing that the observed performance gains do not come from random noise, and we further complement these results with theoretical and empirical analyses. For Reviewer wx4f’s confusion about distilling plans from a strong model, we conduct additional experiments to clarify this design choice. We sincerely appreciate all reviewers for their constructive comments, which have helped us significantly improve the quality of the paper.

Best wishes and regards,

All authors of Submission 19425

---

### Note · Authors · 2026-01-06

I have read and agree with the venue's withdrawal policy on behalf of myself and my co-authors.